# INFERENCE-TIME SCALING OF DISCRETE DIFFUSION MODELS VIA IMPORTANCE WEIGHTING AND OPTIMAL PROPOSAL DESIGN

**Zijing Ou,**[*] **Chinmay Pani,**[*] **Yingzhen Li**
Imperial College London
z.ou22@imperial.ac.uk

## ABSTRACT

Discrete diffusion models have become highly effective across various domains. However, real-world applications often require the generative process to adhere to certain constraints. To this end, we propose a Sequential Monte Carlo (SMC) framework that enables scalable inference-time control of discrete diffusion models through principled importance weighting and optimal proposal construction. Specifically, our approach derives tractable importance weights for a range of intermediate targets and characterises the optimal proposal, for which we develop two practical approximations: a first-order gradient-based approximation and an amortised proposal trained to minimise the log-variance of the importance weights. Empirical results across synthetic tasks, language modelling, biology design, and text-to-image generation demonstrate that our framework enhances controllability and sample quality, highlighting the effectiveness of SMC as a versatile recipe for scaling discrete diffusion models at inference time.

## 1 INTRODUCTION

Diffusion models (Sohl-Dickstein et al., 2015; Ho et al., 2020; Song et al., 2020) have achieved remarkable success across various domains, from image synthesis (Rombach et al., 2022; Esser et al., 2024) to scientific applications (Hoogeboom et al., 2022; Watson et al., 2023). Recently, advances in discrete diffusion models (Austin et al., 2021; Sahoo et al., 2024; Shi et al., 2024) have established them as a powerful approach for modelling discrete data, notably in tasks such as language modelling (Nie et al., 2025; Zhang et al., 2025a) and code generation (Gat et al., 2024; Gong et al., 2025).

Despite their impressive capabilities, pretrained diffusion models often need to generate samples that meet specific downstream constraints. For example, text-to-image generation requires images aligned with human preferences (Black et al., 2023; Fan et al., 2023; Uehara et al., 2024), while protein generation demands stability or desired binding affinity (Verkuil et al., 2022; Uehara et al., 2025b). To address this challenge, existing approaches mainly fall into two categories: i) fine-tuning and ii) guidance methods. Fine-tuning methods, including techniques such as steering (Rector-Brooks et al., 2024), reinforcement learning (Zekri & Boullé, 2025), and direct backpropagation (Wang et al., 2024), have demonstrated promising results. Nevertheless, these methods often suffer from reward over-optimisation, which can compromise sample quality and diversity. On the other hand, guidance and sampling methods (Li et al., 2024; Gruver et al., 2023; Nisonoff et al., 2024; Guo et al., 2024; Uehara et al., 2025a) provide training-free alternatives that are easier to deploy, but they often suffer from reward under-optimisation. This limits their ability to enforce correct alignment, resulting in outputs that may not fully meet complex constraints.

In this paper, with a primary focus on discrete diffusion models, we propose a Sequential Monte Carlo (SMC) (Del Moral et al., 2006) framework for test-time inference. By leveraging SMC, an asymptotically unbiased sampler, our approach enables test-time scaling, effectively addressing the over-optimisation issues commonly encountered by fine-tuning methods. Moreover, we propose a learnable amortised proposal to approximate the optimal SMC proposal, which mitigates the

---

[*]Equal contribution.

under-optimisation problems often associated with guidance-based methods, thereby improving both scalability and efficacy during inference. In summary, our contributions include:

- We propose a simple SMC framework for discrete diffusion models. By leveraging tractable importance weights, we show that SMC provides a general recipe for test-time scaling, enhancing classifier-free guidance and enabling effective reward alignment.

- We propose two approximately optimal proposals: a first-order approximation and a learnable amortised proposal. The latter is optimised by minimising the log-variance of importance weights, leading to substantial improvements in the effectiveness of SMC.

- We demonstrate the versatility of the proposed approach across a broad range of applications, including language modelling, biology design, and text-to-image generation, highlighting its ability to consistently improve performance and generalise across diverse domains.

## 2 BACKGROUND

We first introduce the main preliminaries: discrete diffusion models and Sequential Monte Carlo.

### 2.1 DISCRETE DIFFUSION MODELS

Discrete diffusion models (Austin et al., 2021) define a forward nosing process that interpolates between the original data distribution and a fixed prior $v \in \Delta^V$ on the $V$-simplex:

$$p(x_t|x_0) = \text{Cat}(x_t; \alpha_t x_0 + (1 - \alpha_t)v), \qquad (1)$$

where $\alpha_t$ is a monotonically decreasing schedule from 1 to 0 such that $x_T \sim \text{Cat}(v)$. Masked diffusion models (Sahoo et al., 2024; Shi et al., 2024) are a special case that use a mask token $[\text{m}]$ as the prior, with the induced posterior taking the form of

$$p(x_{t-1}|x_t, x_0) = \begin{cases} \text{Cat}(x_{t-1}; x_t) & x_t \neq [\text{m}], \\ \text{Cat}\left(x_{t-1}; \frac{(1-\alpha_{t-1})[\text{m}] + (\alpha_{t-1} - \alpha_t)x_0}{1 - \alpha_t}\right) & x_t = [\text{m}] \end{cases} \qquad (2)$$

Since $x_0$ is not available during inference, the reverse unmasking process is parametrised as $p_\theta(x_{t-1}|x_t) = p(x_{t-1}|x_t, \mu_\theta(x_t))$, where $\mu_\theta(x_t)$ is a denoising model that predicts the clean data $x_0$. The model is trained by minimising the cross-entropy loss

$$\mathcal{L}(x_0; \theta) = \sum_{t=1}^{T} \frac{\alpha'_t}{1 - \alpha_t} \mathbb{E}_{p(x_t|x_0)}[-\log(x_0^T \mu_\theta(x_t))] \, dt, \qquad (3)$$

which is equivalent, if $T \to \infty$, to the negative evidence lower bound of the log-likelihood $\log p_\theta(x_0)$.

### 2.2 IMPORTANCE SAMPLING AND SEQUENTIAL MONTE CARLO

Consider the Monte Carlo integration problem $\mathbb{E}_{\pi(x_t)}[\delta(x_t)]$, where sampling from the target distribution $\pi$ is intractable. Importance sampling (Robert et al., 1999) alleviates this issue by introducing a proposal distribution $q$, allowing the expectation to be rewritten as

$$\mathbb{E}_{\pi(x_t)}[\delta(x_t)] = \mathbb{E}_{q(x_{t:T})}\left[\frac{\pi(x_{t:T})}{q(x_{t:T})}\delta(x_t)\right] \approx \frac{1}{N} \sum_{i=1}^{N} \frac{\pi(x_{t:T}^{(i)})}{q(x_{t:T}^{(i)})}\delta(x_t^{(i)}), \quad x_{t:T}^{(i)} \sim q(x_{t:T}). \qquad (4)$$

While conceptually simple, importance sampling often suffers from high variance. To address this limitation, Sequential Monte Carlo (SMC) (Del Moral et al., 2006) extends importance sampling by incorporating resampling and sequential weighting strategies across the path, thereby reducing variance in practice. In SMC, a key intuition is the recursive formulation of the importance weight

$$w_{t-1}(x_{t-1:T}^{(i)}) \triangleq \frac{\pi(x_{t-1:T}^{(i)})}{q(x_{t-1:T}^{(i)})} = \frac{\pi(x_{t-1}^{(i)}|x_{t:T}^{(i)})\pi(x_{t:T}^{(i)})}{q(x_{t-1}^{(i)}|x_{t:T}^{(i)})q(x_{t:T}^{(i)})} = \frac{\pi(x_{t-1}^{(i)})}{\pi(x_t^{(i)})} \frac{\gamma(x_t^{(i)}|x_{t-1}^{(i)})}{q(x_{t-1}^{(i)}|x_t^{(i)})}w_t(x_{t:T}^{(i)}), \qquad (5)$$

where we leverage the Markovian assumption that $\pi(x_{t:T}^{(i)}) = \pi(x_t^{(i)})\prod_{k=t}^{T-1}\gamma(x_{t+1}^{(i)}|x_t^{(i)})$ for arbitrary forward kernel $\gamma$ and thus $\pi(x_{t-1}^{(i)}|x_t^{(i)}) = \pi(x_{t-1}^{(i)})\gamma(x_t^{(i)}|x_{t-1}^{(i)})/\pi(x_t^{(i)})$. The recursion of

importance weight underlies the iterative procedure of SMC. Concretely, The procedure initialises begins by $N$ particles $x_T^{(i)} \sim q(x_T)$ with weights $w_T^{(i)} \leftarrow \pi(x_T^{(i)})/q(x_T^{(i)})$. For each step $t = T, \ldots, 1$ and particles $i = 1, \ldots, N$, SMC proceeds as follows: i) resample ancestor according to the weights $\{w_t^{(i)}\}_{i=1}^N$; ii) propagate new particles via $x_{t-1}^{(i)} \sim q(x_{t-1}|x_t)$; and iii) updating the weights as $w_{t-1}^{(i)} \leftarrow [\pi(x_{t-1}^{(i)})\pi(x_t^{(i)}|x_{t-1}^{(i)})]/[\pi(x_t^{(i)})q(x_{t-1}^{(i)}|x_t^{(i)})]$. The resulting collection of weighted particles provides an asymptotically consistent approximation of the intermediate target distribution $\pi(x_t)$.

## 3 SEQUENCE MONTE CARLO FOR DISCRETE DIFFUSION MODELS

Given a pretrained discrete diffusion model $p_\theta(x_t)$, we consider sampling from modified target distributions that enable inference-time control. These targets include: i) product distributions, a general form underlying classifier free guidance (Ho & Salimans, 2022), defined as $\pi(x_t) \propto p_{\theta_1}^\alpha(x_t)p_{\theta_2}^\beta(x_t)$; and ii) reward-tilting distributions, expressed as $\pi(x_t) \propto p_\theta(x_t)\exp(r(x_t))$. In the following section, we introduce how to construct tractable importance weights by carefully selecting forward kernels $\gamma(x_t|x_{t-1})$, and show their connection to existing SMC formulations for continuous-time discrete diffusion models. We then discuss the choice of proposal distributions, which play a central role in balancing variance reduction with computational efficiency.

### 3.1 IMPORTANCE WEIGHT: TRACTABILITY WITH PRETRAINED DIFFUSION MODELS

To perform SMC, one must evaluate the importance weight from Equation (5) at each step $t$. While the forward kernel $\gamma(x_t|x_{t-1})$ and the proposal $q(x_{t-1}|x_t)$ can be chosen flexibly, the ratio of intermediate targets $\frac{\pi(x_{t-1})}{\pi(x_t)}$ is generally intractable in diffusion models. With a well-trained diffusion model $p_\theta$, however, this ratio can be approximated via detailed balance $\frac{p_\theta(x_{t-1})}{p_\theta(x_t)} \approx \frac{p_\theta(x_{t-1}|x_t)}{p(x_t|x_{t-1})}$, where $p(x_t|x_{t-1})$ denotes the forward noising process and $p_\theta(x_{t-1}|x_t)$ is the learned reverse counterpart. Under this approximation, the importance weight for the product target takes the form

$$\text{product:} \quad \frac{p_{\theta_1}^\alpha(x_{t-1}|x_t)p_{\theta_2}^\beta(x_{t-1}|x_t)}{p_1^\alpha(x_t|x_{t-1})p_2^\beta(x_t|x_{t-1})} \frac{\gamma(x_t|x_{t-1})}{q(x_{t-1}|x_t)}. \tag{6}$$

Although tractable, this weight inevitably introduces approximation error unless the reverse model is perfectly trained, due to the mismatch between the forward and backward processes. In contrast, for the reward-tilting, the error can be eliminated by setting $\gamma(x_t|x_{t-1}) = p(x_t|x_{t-1})$, yielding

$$\text{reward-tilting:} \quad \frac{\exp(r(x_{t-1}))}{\exp(r(x_t))} \frac{p_\theta(x_{t-1}|x_t)}{p(x_t|x_{t-1})} \frac{\gamma(x_t|x_{t-1})}{q(x_{t-1}|x_t)} = \frac{\exp(r(x_{t-1}))}{\exp(r(x_t))} \frac{p_\theta(x_{t-1}|x_t)}{q(x_{t-1}|x_t)} \tag{7}$$

It is noteworthy that this cancellation is not applicable to the product distributions, since the normalising constant of $\gamma$ can not be cancelled even if we choose $\gamma \propto p_1^\alpha(x_t|x_{t-1})p_2^\beta(x_t|x_{t-1})$. Nevertheless, as illustrated in Figure 1, SMC with these tractable importance weights performs well across both two settings on 2D toy examples. Moreover, although we primarily focus on discrete-time diffusion, the proposed SMC method can be extended seamlessly to the continuous-time setting, as established in the following proposition.

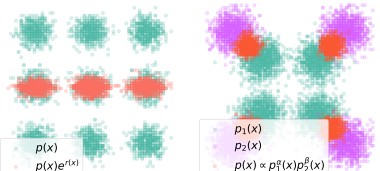

Figure 1: SMC results for the reward-tilting and product target distributions.

**Proposition 1** (SMC for Continuous-Time Discrete Diffusion). *Let $R_t$ be the rate matrix generating the forward transition kernel $\gamma(x_t|x_{t-\Delta t})$, and $\hat{R}_t$ be its counterpart associated with the backward proposal kernel $q(x_{t-\Delta t}|x_t)$, where $\Delta t \to 0$ is the infinitesimal time increment. Then, the importance weight at time $t$ is given by $w_t = \int_1^t -\partial_s \log \pi(x_s) + \sum_{y_s} R_s(x_s, y_s)\frac{\pi(y_s)}{\pi(x_s)} \, ds$, if the forward kernel $\gamma$ is chosen such that the rate matrices satisfy detailed balance $\hat{R}_t(x_t, y_t)\pi(x_t) = R_t(y_t, x_t)\pi(y_t)$.*

Proposition 1 coincides with the importance weight used in Holderrieth et al. (2025), which focus on sampling scenarios where the intermediate target $\pi$ is tractable up to an unnormalized constant. In contrast, our setting concerns test-time control of pretrained diffusion models, where $\pi$ is intractable. In Appendix B.2, we further connect our result to Lee et al. (2025), who also study reward tilting, but we provide a derivation from the perspective of discrete-time diffusion.

---

**Algorithm 1** Training Optimal Proposal

1: Rollout trajectory $\{x_t\}_t$ with $q_{\text{ref}}(x_{t-1}|x_t)$
2: Compute gradient with Equation (10)
   $$g_\phi, g_\psi \leftarrow \frac{1}{T}\nabla_{\phi,\psi}\sum_t \mathcal{L}_{\phi,\psi}(x_{t-1}, x_t)$$
3: Update $\phi, \psi$ using $g_\phi, g_\psi$
   $$\phi \leftarrow \phi - \eta g_\phi, \quad \psi \leftarrow \psi - \eta g_\psi$$

---

**Algorithm 2** SMC Sampling Procedure

1: Propose particles $x_{t-1}^{(i)} \sim q(x_{t-1}|x_t)$
2: Compute importance weight with Equation (5)
   $$w_{t-1}^{(i)} = \frac{\pi(x_{t-1})}{\pi(x_t)}\frac{\gamma(x_t|x_{t-1})}{q(x_{t-1}|x_t)}, \tilde{w}_{t-1}^{(i)} = \frac{w_{t-1}^{(i)}}{\sum_i w_{t-1}^{(i)}}$$
3: Resample $x_{t-1}^{(i)} \sim \text{Multinormial}(x_{t-1}^{(i)}; \tilde{w}_{t1}^{(i)})$

---

## 3.2 CHOICES OF PROPOSAL: THE WAY TO REDUCE VARIANCE

While the proposal $q(x_{t-1}|x_t)$ offers substantial flexibility, the statistical efficiency of SMC is highly sensitive to its choice: suboptimal proposals induce high-variance importance weights, which in turn precipitate particle degeneracy and hinder adequate exploration of the state space (Del Moral et al., 2006). Conversely, an appropriately constructed proposal substantially mitigates weight variance, thereby enhancing the effective sample size and ensuring stability of the inference procedure. The following proposition characterises the minimum variance choice of proposal:

**Proposition 2** (Locally Optimal Proposal). *Given the incremental importance weight as in Equation (5) $w_{t-1}(x_{t-1}, x_t) = \frac{\pi(x_{t-1})\gamma(x_t|x_{t-1})}{\pi(x_t)q(x_{t-1}|x_t)}$, the proposal distribution that minimises the variance of $w_{t-1}$, often referred to as the* locally optimal proposal, *is $q(x_{t-1}|x_t) \propto \pi(x_{t-1})\gamma(x_t|x_{t-1})$.*

Building on Proposition 2, one readily verifies that the optimal proposal distribution for the case of product target is tractable (see the remark in Appendix A.2 for discussion), under the choices of forward kernels $\gamma \propto p_1^\alpha(x_t|x_{t-1})p_2^\beta(x_t|x_{t-1})$. In contrast, for the reward-tilting, the locally optimal proposal takes the form $q \propto \exp(r(x_{t-1}))p_\theta(x_{t-1}|x_t)$, which is generally computationally infeasible due to the inaccessibility of the corresponding normalising constant. Consequently, practical implementations must resort to approximations that balance variance reduction with computational feasibility. In what follows, we introduce two approximation strategies tailored to the reward-tilting setting: i) a gradient-based method to achieve first-order approximation, and ii) a neural proposal trained to minimise the log-variance of the importance weight.

### 3.2.1 APPROXIMATED OPTIMAL PROPOSAL VIA FIRST-ORDER APPROXIMATION

In reward tilting, evaluating the locally optimal proposal requires computing the normalising constant $Z = \sum_{x_{t-1}} \exp(r(x_{t-1}))p_\theta(x_{t-1}|x_t)$. This computation entails $\mathcal{O}(|\mathcal{X}|)$ forward pass through the reward model at each denoising step, which significantly slows down the generation speed, rendering the method impractical for large discrete state spaces. To mitigate this issue, we adopt the approach of Grathwohl et al. (2021); Zhang et al. (2022), treating $r(x_t)$ as a function defined over continuous real-valued inputs, while evaluating it on the discretised domain of interest. This allows us to apply a first-order Taylor expansion to approximate the reward: $r(x_{t-1}) \approx r(x_t) + (x_{t-1} - x_t)^T\nabla_x r(x_t)$, which in turn yields a first-order approximation to the locally optimal proposal:

$$q(x_{t-1}|x_t) \propto p_\theta(x_{t-1}|x_t)\exp(x_{t-1}^T\nabla_x r(x_t)). \tag{8}$$

This approximation improves computational efficiency by requiring the reward function $r$ to be evaluated and differentiated only once at $x_t$, instead of repeatedly across all states. Nevertheless, it assumes differentiable rewards and remains costly when the reward model is large. Motivated by Richter et al. (2020); Richter & Berner (2023) and the amortisation technique in variational inference (Dayan et al., 1995; Kingma & Welling, 2013), we propose learning an amortised network that approximates the optimal proposal, resulting in a transition kernel that directly transports between successive intermediate targets as in Matthews et al. (2022). This reduces computation to a single network evaluation, thereby significantly enhancing the efficiency of SMC.

### 3.2.2 AMORTISED OPTIMAL PROPOSAL VIA LOG-VARIANCE MINIMISATION

To train a network $q_\phi$ to approximate the locally optimal proposal, a natural approach is to minimise the log-variance of the importance weight:

$$\min_\phi \mathbb{V}_{q_{\text{ref}}(x_{0:T})}\left[\sum_t \log \frac{\exp(r(x_{t-1}))}{\exp(r(x_t))}\frac{p_\theta(x_{t-1}|x_t)}{q_\phi(x_{t-1}|x_t)}\right] \triangleq \mathcal{L}_{\text{log-var}}(\phi) \tag{9}$$

where $q_{\text{ref}}$ is an arbitrary reference distribution that has the same support as $p_\theta$ and $q_\phi$. The following corollary establishes the validity of the proposed log-variance objective.

**Corollary 1.** *The locally optimal proposal $q^* \propto \pi(x_{t-1})p_\theta(x_{t-1}|x_t)$ that achieves the minimum variance of the important weight $\mathbb{V}_q\left[\frac{\pi(x_{t-1})\gamma(x_t|x_{t-1})}{\pi(x_t)q(x_{t-1}|x_t)}\right]$ is unique.*

Although conceptually simple, naive Monte Carlo estimation of $\mathcal{L}_{\text{log-var}}$ suffers from high variance and computational cost. To alleviate these issues, we introduce an auxiliary network $F_\psi : \mathbb{R} \to \mathbb{R}$, parameterised by $\psi$, that estimates the mean of the log-weight. This yields the refined objective:

$$\mathcal{L}(\phi, \psi) = \mathbb{E}_{t,q_{\text{ref}}(x_{t-1},x_t)} \left|\log \frac{\exp(r(x_{t-1}))}{\exp(r(x_t))} \frac{p_\theta(x_{t-1}|x_t)}{q_\phi(x_{t-1}|x_t)} - F_\psi(t)\right|^2, \tag{10}$$

which provably upper bounds the log-variance loss. To be specific, the following proposition holds:

**Proposition 3.** *For any reference distribution $q_{\text{ref}}$, we have $\mathcal{L}_{\text{log-var}}(\phi) \leq T^2 \mathcal{L}(\phi, \psi)$. Moreover, the minimiser of $\mathcal{L}$ is unique and attains its optimum when $q_\phi \propto \exp(r(x_{t-1}))p_\theta(x_{t-1}|x_t)$.*

We outline the training and sampling procedures in Algorithms 1 and 2. For clarity, we designate $\text{SMC}_{\text{base}}$, $\text{SMC}_{\text{grad}}$, and $\text{SMC}_{\text{amot}}$ to denote, respectively, the variants employing the pretrained diffusion proposal, the first-order approximated proposal, and the learned amortised proposal. We further denote the first-order approximated and amortised proposals by $\text{Prop}_{\text{grad}}$ and $\text{Prop}_{\text{amot}}$, which coincide with their corresponding SMC methods when restricted to a single particle.

### 3.3 Sequential Monte Carlo Recipe: Practical Implementation

Building on the theoretical characterisation of optimal proposals, we next present a practical SMC recipe. An essential ingredient of the proposed framework is the introduction of a twisted intermediate target for reward-tilting: $\pi(x_t) \propto p_\theta(x_t)\exp\left(\frac{\lambda_t}{\alpha}r(x_t)\right)$, where $\alpha > 0$ is a KL-regularisation coefficient. This construction is motivated by the following identity

$$\pi = \underset{\pi}{\arg\max} \, \mathbb{E}_\pi[r(x_t)] - \alpha\mathbb{KL}(\pi\|p_\theta) \propto p_\theta(x_t)\exp\left(\frac{r(x_t)}{\alpha}\right). \tag{11}$$

Here, $\lambda_t \in [0, 1]$ acts a temperature parameter that smoothly interpolates between the prior $p_\theta(x_t)$ ($\lambda_t = 0$) and the fully reward-augmented target ($\lambda_t = 1$). By gradually increasing $\lambda_t$ over denoising steps, the influence of the reward is tempered, thereby improving stability during sampling. In scenarios where the reward is only defined on clean data, following Wu et al. (2023a); Kim et al. (2025), we approximate the optimal intermediate target as

$$\pi(x_t) \propto p_\theta(x_t)\exp\left(\frac{\lambda_t}{\alpha}\hat{r}(x_t)\right), \quad \hat{r}(x_t) = \frac{1}{M}\sum_{m=1}^{M} r(x_0^{(m)}), \quad x_0^{(m)} \sim p_\theta(x_0|x_t). \tag{12}$$

However, categorical sampling from $p_\theta$ renders $\hat{r}(x_t)$ non-differentiable w.r.t. $x_t$. To resolve it, we employ the reparameterisation trick with Gumbel-Softmax (Jang et al., 2016) to enable differentiability (see Appendix C.2 for details), thereby making the approximated proposal applicable as in Equation (8). For completeness, Appendix C.3 provides a further discussion of the computation of importance weights in Equation (7) under low-confidence sampling, where the ratio $\frac{p_\theta(x_{t-1}|x_t)}{q(x_{t-1}|x_t)}$ is not explicitly tractable. This extension ensures that the proposed SMC algorithm remains suitable for recent state-of-the-art discrete diffusion models for language modelling (Nie et al., 2025) and text-to-image generation (Bai et al., 2024), where low-confidence sampling (Chang et al., 2022) is commonly used.

## 4 Experiments

To support our theoretical discussion, we first showcase the effectiveness of the proposed methods through a synthetic experiment. We then evaluate it across a wide range of applications, including language modelling, biological design, and text-to-image generation. Detailed experimental settings and additional results are provided in Appendix D.

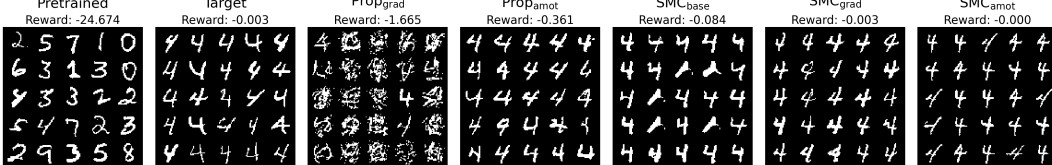

Figure 2: Comparisons on reward-tilted discreteised MoGs. We consider the reward function as $r(X, Y) = -\hat{X}^2/100 - \hat{Y}^2$, where $\hat{X} = 12(X/63 - 1/2)$ and $\hat{Y} = 12(Y/63 - 1/2)$.

Figure 3: Comparisons on reward-tiled binary MNIST. We train a classier $p_{\text{clf}}(y|x)$ on the clean data, and the reward is given by $r(x) = \log p_{\text{clf}}(y_{\text{target}}|x)$, where $y_{\text{target}}$ denotes the target digit.

## 4.1 SYNTHETIC EXPERIEMNTS

We begin with the empirical evaluation with two synthetic experiments: binary MNIST and a two-dimensional discretised mixture of Gaussians (MoG), each dimension comprising 64 categorical states. A discrete diffusion model is first pretrained on the clean data. We evaluate the proposed SMC-based reward-tilting variants in comparison with two non-SMC baselines: $\text{Prop}_{\text{grad}}$, corresponding to the first-order approximation in Equation (8), and $\text{Prop}_{\text{amot}}$, which utilises an amortised proposal trained according to the objective in Equation (10). Performance is assessed using the earth mover's distance (EMD), alongside the evaluation of the reward on the generated samples.

The results, shown in Figures 2 and 3, indicate that, compared to the non-SMC baselines, the SMC-based methods achieve superior performance, demonstrating the effectiveness of the proposed approach. Specifically, $\text{SMC}_{\text{amot}}$ attains the highest rewards and lowest EMD, though at the cost of reduced sample diversity, likely due to reward over-optimisation. In contrast, $\text{SMC}_{\text{grad}}$ maintains comparable sample quality while preserving high diversity, highlighting the effectiveness of the proposed approximated optimal proposal. Furthermore, $\text{Prop}_{\text{amot}}$ significantly outperforms $\text{Prop}_{\text{grad}}$, underscoring the benefit of the log-variance minimisation objective. We further demonstrate the reward curves over training in Figures 7a and 7b, which shows that the proposed method can achieve stable reward convergence, confirming the efficiency and robustness in learning optimal proposal.

## 4.2 LANGUAGE MODELLING

We further evaluate our approach on language modelling, focusing on toxic text generation (Singhal et al., 2025), an undesirable behaviour of language models, where the pretrained model MDLM (Sahoo et al., 2024) produces only $0.8\%$ of samples flagged as toxic. To assess sample quality, we use four metrics: i) Toxic, based on the same reward model applied during inference (Logacheva et al., 2022); ii) Toxic (Holdout), measured by a holdout toxicity classifier trained on a multilingual mixture of datasets (Dementieva et al., 2024); iii) generative perplexity with GPT2-XL (Radford et al., 2019); and iv) distinct uni/bi/trigrams (Dist-1/2/3). The first two metrics evaluate alignment with the reward, while the latter two measure semantic quality and diversity.

Following Han et al. (2022), we generate sequences of length 100 with 100 denoising steps condition on the given starting prompts, and report results averaged over 300 independent runs corresponding to 15 prompts with 20 generations per prompt. The results are summarised in Table 1, with an extended version provided in Table 4. We observe that SMC with proposals closer to the optimal achieves better performance on the toxicity metrics, reflecting stronger alignment with the reward model. Among the non-SMC baselines, $\text{Prop}_{\text{amot}}$ yields the best performance, highlighting the effectiveness of the log-variance minimisation objective. To further assess its effect, we plot the training dynamics in Figure 7c, which shows the reward steadily improving as training progresses. Notably, although the learned proposal sacrifices a small degree of performance on perplexity and diversity, we demonstrate

Table 1: The results of toxic text generation. We use a widely adopted toxicity classifier as the reward (Logacheva et al., 2022), while the pretrained language model is MDLM (Sahoo et al., 2024).

| # Particles | Method | Toxic ↑ | Toxic (Holdout) ↑ | PPL (GPT2-XL) ↓ | Dist-1/2/3 ↑ |
|---|---|---|---|---|---|
| N = 1 | Pretrained | 0.8% | 5.2% | 121.1 | 56/92/96 |
| | $\text{Prop}_{\text{grad}}$ | 58.0% | 58.3% | 216.7 | 58/93/96 |
| | $\text{Prop}_{\text{amot}}$ | 63.7% | 75.7% | 131.9 | 53/89/94 |
| N = 8 | BoN | 6.3% | 16.7% | 127.4 | 56/91/96 |
| | $\text{SMC}_{\text{base}}$ | 26.7% | 40.0% | 132.3 | 57/92/96 |
| | $\text{SMC}_{\text{grad}}$ | 95.0% | 86.3% | 132.1 | 57/92/96 |
| | $\text{SMC}_{\text{amot}}$ | 100.0% | 99.7% | 147.6 | 44/81/91 |

Figure 4: Results of DNA sequence design. Both the pretrained discrete diffusion model and the reward models are adopted from Wang et al. (2024).

in Figure 19 that it consistently generates coherent and semantically meaningful sequences, indicating that alignment improvements need not come at the expense of sample quality.

### 4.3 BIOLOGY DESIGN

In this experiment, we evaluate our method on DNA sequence design. Specifically, we adopt the pretrained model and the reward model from Wang et al. (2024), which are trained on ∼700k DNA sequences. To evaluate the performance, we consider five metrics: i) predicted activity (*Pred-Activity*); ii) chromatin accessibility classification accuracy (*ATAC-Acc*); iii) 3-mer Pearson correlation with dataset sequences (*3-mer Corr*); iv) JASPAR motif frequency correlation (*JASPAR Corr*); and v) approximate log-likelihood under the pretrained model (*App-Log-Lik*). For further details on these evaluation metrics, we refer the reader to Wang et al. (2024).

As shown in Figure 4, the performance improves consistently with an increasing number of particles, suggesting that SMC benefits from a larger particle set by providing a more accurate approximation of the target distribution. Compared to $\text{SMC}_{\text{base}}$, $\text{SMC}_{\text{amot}}$ achieves higher *Pred-Activity* and *ATAC-Acc*, while performing slightly worse on the other three metrics. This can be attributed to the more mode-seeking behaviour of their proposals, which emphasises high-probability regions at the expense of overall diversity. Nonetheless, we observe that $\text{SMC}_{\text{amot}}$ with larger particle sets attains more favourable overall performance, indicating that a learnable amortised proposal can effectively leverage the flexibility of SMC to balance quality and diversity. In Appendix D.2.4, we present additional comparisons with baseline methods, further demonstrating the effectiveness of our approach.

### 4.4 IMAGE GENERATION

In this section, we evaluate our method on image generation. We begin by demonstrating that SMC yields improvements over classifier-free guidance, which can be viewed as a special case of the product target in Equation (6). Subsequently, we present large-scale experiments to illustrate the applicability of the proposed methods to text-to-image generation at scale.

**Improving CFG with MaskGit (Chang et al., 2022).** Given a pretrained diffusion model $p_\theta$, classifier-free guidance (CFG) generates samples according to $p_\theta(x_{t-1}|x_t, c)^\alpha p_\theta(x_{t-1}|x_t)^{1-\alpha}$, where $\alpha$ is the CFG coefficient. CFG has been shown to enhance sample quality substantially (Ho & Salimans, 2022). By incorporating the importance weight defined in Equation (6), we can further improve CFG within the proposed SMC framework.

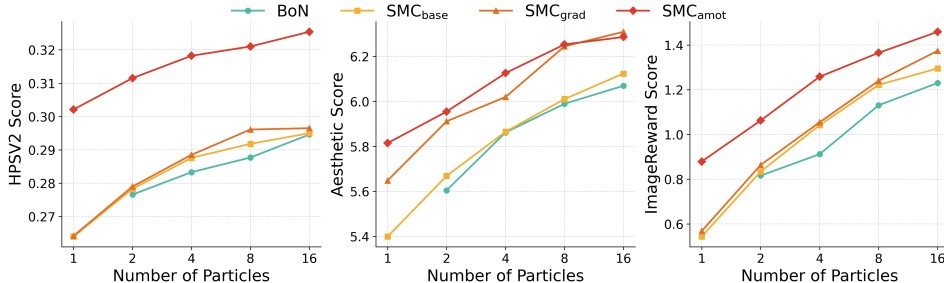

Figure 5: The results of text-to-image generation across different reward models.

Specifically, we perform experiments with MaskGit (Chang et al., 2022) trained on ImageNet256 (Deng et al., 2009). To ensure tractable importance weight computation, we adopt the ReMDM sampling scheme (Wang et al., 2025) instead of the low-confidence sampling strategy from (Chang et al., 2022) (see Tables 9 to 11 for a comparison). The result is presented in Table 2. It shows that with fewer denoising steps, increasing the number of particles leads to a substantial improvement in sample quality, as measured by FID and Inception Score (IS) with $50,000$ generated images, thereby demonstrating the effectiveness of the proposed SMC method. However, as the number of denoising steps increases, the benefit of using more particles diminishes. This can be attributed to the role of SMC in correcting sampling inaccuracies: with sufficient denoising steps, the sampling process itself becomes accurate enough, leaving limited room for further improvement through additional particles. In addition, we report results with different CFG coefficients in Tables 7 and 8. Interestingly, for larger CFG coefficients, increasing the particle count tends to decrease FID while increasing IS. This behaviour is expected, since stronger CFG reduces sample diversity. With more accurate sampling under SMC, this reduction in diversity becomes more apparent, leading to lower FID but higher IS.

Table 2: Comparisons of different numbers of particles with CFG=1.25 on ImageNet256.

| | FiD ↓ | | | IS ↑ | | |
|---|---|---|---|---|---|---|
| # steps | 8 | 16 | 32 | 8 | 16 | 32 |
| N = 1 | 24.64 | 14.94 | 12.02 | 62.8 | 90.7 | 107.5 |
| N = 2 | 21.08 | 12.55 | 10.26 | 74.0 | 106.6 | 126.2 |
| N = 4 | 18.08 | 10.92 | 9.29 | 87.4 | 123.5 | 146.5 |
| N = 8 | 16.26 | 9.93 | 8.98 | 96.4 | 139.4 | 159.8 |
| N = 16 | 14.56 | 9.59 | 8.76 | 107.4 | 149.4 | 170.7 |

**Improving Text-to-Image Generation with Meissonic (Bai et al., 2024).** We evaluate the scalability of the proposed methods on text-to-image generation using Meissonic (Bai et al., 2024) as the base model. Our experiments consider three text–image alignment rewards: Human Preference Score (HPSv2) (Wu et al., 2023b), Aesthetic Score (LAION, 2024), and ImageReward (Xu et al., 2023). For the prompt distributions, we use photo and painting prompts from the Human Preference Dataset (HPDv2) (Wu et al., 2023b) for HPSv2, the DrawBench prompt set for ImageReward, and a curated set of 45 simple animal prompts for Aesthetic Score, follwoing Black et al. (2023).

The results are shown in Figure 5. We observe that performance consistently improves with an increasing number of particles, and SMC$_{amot}$ outperforms all other methods, which highlights the benefit of the proposed SMC framework. In Figure 6, we visualise the alignment dynamics for the HPSv2 task, showing that the generated images progressively align more faithfully with the prompts, thereby validating the effectiveness of the proposed log-variance minimisation objective. Furthermore, Figures 7e to 7g present the convergence of the reward during training, and further qualitative examples in Appendix D.2.7 collectively reinforce the validity of our approach.

## 5 RELATED WORK

**Discrete Diffusion Models.** Discrete diffusion models (DDMs) were originally introduced in Austin et al. (2021); Sun et al. (2022); Campbell et al. (2022), grounded in the framework of continuous-time Markov chains (Norris, 1998). More recently, masked diffusion models (MDMs) (Lou et al., 2023; Shi et al., 2024; Sahoo et al., 2024), a special case of DDMs, have shown strong performance in language modelling (Zhang et al., 2025a; Nie et al., 2025). In addition, MDMs have achieved promising results in math reasoning (Zhao et al., 2025), image synthesis (Bai et al., 2024), code planning (Gat et al., 2024; Gong et al., 2025), and biological sequence generation (Campbell et al., 2024), yielding

— A helmet-wearing monkey skating. —>

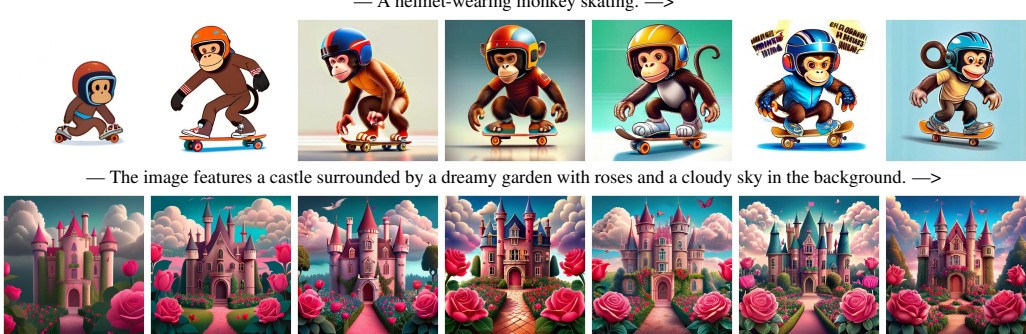

— The image features a castle surrounded by a dreamy garden with roses and a cloudy sky in the background. —>

Figure 6: Visualisation of alignment dynamics over the training progress, with images generated every 100 training steps. The generated images become more faithful to the text prompt.

performance comparable to continuous diffusion (Rombach et al., 2022) and autoregressive models (Radford et al., 2019). In contrast to these approaches, which primarily study large-scale pretraining, our work focuses on test-time inference and post-training alignment (Uehara et al., 2025b), where access to training data is not available.

**Test-time Alignment of Discrete Diffusion Models.** Existing alignment methods mainly fall into two categories: classifier guidance (Dhariwal & Nichol, 2021) and RL-based fine-tuning (Black et al., 2023). Although the score is ill-defined in discrete distributions, several works (Vignac et al., 2022; Nisonoff et al., 2024; Schiff et al., 2024) employ a first-order approximation to the target distribution, which resembles the insight underlying our approximated proposal $\text{Prop}_{\text{grad}}$ in Equation (8). Alternatively, Rout et al. (2025) perform guidance on the embedding space, mitigating the issue of ill-defined gradients. Chen et al. (2025b) introduce reward-free guidance, analogous to classifier-free guidance (Ho & Salimans, 2022) but designed for masked diffusion models. Moreover, Tang et al. (2025) propose tree search guided finetuning, which is related to the searching-based scaling methods on continuous diffusion (Ma et al., 2025; Zhang et al., 2025b; Jain et al., 2025; Ramesh & Mardani, 2025). Beyond guidance approaches, sampling-based techniques have also demonstrated promising performance, including value-based sampling (Li et al., 2024), importance sampling (Guo et al., 2024), and iterative refinement strategies (Uehara et al., 2025a). While training-free and relatively efficient to deploy, these methods often face challenges in scalability and robustness. More recently, RL-based fine-tuning methods (Zekri & Boullé, 2025; Zhao et al., 2025; Gong et al., 2025) have gained significant traction, fueled by the remarkable success of Group Relative Policy Optimisation (GRPO) (Shao et al., 2024) in large language models (Guo et al., 2025). In parallel, steering-based (Rector-Brooks et al., 2024) approaches leveraging GFlowNets (Bengio et al., 2023) and direct backpropagation methods (Wang et al., 2024) have also demonstrated strong potential for test-time alignment. Distinct from these directions, our amortised proposal $\text{Prop}_{\text{amot}}$ introduces an alternative perspective for fine-tuning pretrained discrete diffusion models: it minimises the log-variance of importance weights, a criterion that has been rarely investigated in previous work.

**Sequential Monte Carlo for Generative Modelling.** SMC has emerged as a versatile framework for probabilistic modelling, providing effective tools for sampling and inference across a wide range of applications, including particle filtering (Johansen, 2009), Bayesian experimental design (Ryan et al., 2016), and probabilistic planning (Piché et al., 2018). Most recently, SMC has been combined with diffusion models (Chen et al., 2024; He et al., 2025; Skreta et al., 2025; Wu et al., 2025; Ou et al., 2026), transforming it into a powerful neural sampler capable of drawing from complex Boltzmann distributions. These developments have also extended SMC's reach to discrete domains, as demonstrated by Holderrieth et al. (2025); Lee et al. (2025). Beyond classical sampling tasks, SMC has further expanded to the improvement of generative models at test time. A seminal step in this direction was taken by Zhao et al. (2024), who introduced SMC as a principled probabilistic inference framework for addressing capability and safety challenges in large language models (LLMs). Subsequent works (Feng et al., 2024; Puri et al., 2025) successfully applied this idea to enhance mathematical reasoning in LLMs, while others explored its use in reward-guided adaptation of pretrained diffusion models (Trippe et al., 2022; Wu et al., 2023a; Cardoso et al., 2023; Dou & Song, 2024; Kim et al., 2025; Yoon et al., 2025; Chen et al., 2025a; Ren et al., 2025). Our work is most closely related to Singhal et al. (2025); Dang et al. (2025); Hasan et al. (2025), who employ SMC for test-time alignment of discrete diffusion models. However, their approaches treat pretrained diffusion

models as fixed proposal distributions. By contrast, we take a closer look at the role of proposal choice, systematically investigating its impact and providing empirical evidence for a key insight: proposals that better approximate the optimal, which minimises the variance of importance weights, consistently lead to better performance.

## 6 CONCLUSION

In this paper, we introduced a Sequential Monte Carlo (SMC) framework tailored for discrete diffusion models. By exploiting tractable importance weights, we established SMC as a powerful and principled recipe for test-time scaling. A central insight of our work is that the proposal distribution is crucial for unlocking the full potential of SMC. Building on this observation, we developed two approximately optimal proposals: a first-order approximation and a learnable amortised proposal trained to approximate the optimal proposal by minimising the log-variance of importance weights. Extensive experiments across diverse domains demonstrated the effectiveness and scalability of our approaches. We hope this work inspires future studies on more efficient test-time scaling and post-training alignment strategies for discrete diffusion models.

## 7 ACKNOWLEDGEMENTS

ZO is supported by the Lee Family Scholarship. We would like to thank the anonymous reviewers for their constructive and valuable suggestions. ZO also thanks Ruixiang Zhang for insightful discussions and for his support with the experimental setup.

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

# Appendix for "Inference-Time Scaling of Discrete Diffusion Models via Importance Weighting and Optimal Proposal Design"

## CONTENTS

## A  ABSTRACT PROOF AND DERIVATIONS

### A.1  A BRIEF RECAP OF SMC

In this section, we provide a brief overview of Sequential Monte Carlo (SMC). For a target distribution $\pi(x_t)$, we consider the problem of estimating the expectation of a test function $\delta$, namely $\mathbb{E}_{\pi(x_t)}[\delta(x_t)]$. When $\delta(\cdot)$ is taken to be the Dirac delta function, estimating this expectation reduces to constructing an empirical approximation of the distribution $\pi(x_t)$.

To estimate the expectation, importance sampling introduces a proposal distribution $q$, which is easy to sample from, and proposes an estimator as follows

$$\mathbb{E}_{\pi(x_t)}[\delta(x_t)] = \mathbb{E}_{q(x_{t:T})}\left[\frac{\pi(x_{t:T})}{q(x_{t:T})}\delta(x_t)\right] \approx \sum_{i=1}^{N} w_t^{(i)}\delta(x_t^{(i)}), \text{ where } w_t^{(i)} = \frac{\pi(x_{t:T}^{(i)})}{q(x_{t:T}^{(i)})}, x_{t:T}^{(i)} \sim q(x_{t:T})$$

The key ingredients of SMC are the target distribution $\pi(x_{t:T})$ and the proposal distribution $q(x_{t:T})$. Here we consider the target distribution as a Markovian model associated with a sequence of forward transition kernels $\gamma$: $\pi(x_{t:T}) = \pi(x_t)\prod_{s=t}^{T-1}\gamma(x_{t+1}|x_t)$; and the proposal distribution as

$q(x_{t:T}) = \pi(x_T) \prod_{s=t}^{T-1} q(x_t|x_{t+1})$. Substituting these into the importrance weights gives

$$
\begin{aligned}
w_t &= \frac{\pi(x_t) \prod_{s=t}^{T-1} \gamma(x_{t+1}|x_t)}{\pi(x_T) \prod_{s=t}^{T-1} q(x_t|x_{t+1})} \\
&= \frac{\pi(x_t)\gamma(x_{t+1}|x_t)}{\pi(x_{t+1})q(x_t|x_{t+1})} \frac{\pi(x_{t+1}) \prod_{s=t+1}^{T-1} \gamma(x_{t+1}|x_t)}{\pi(x_T) \prod_{s=t+1}^{T-1} q(x_t|x_{t+1})} \\
&= \frac{\pi(x_t)\gamma(x_{t+1}|x_t)}{\pi(x_{t+1})q(x_t|x_{t+1})} w_{t+1}
\end{aligned}
\tag{13}
$$

This recursive structure allows importance weights to be computed incrementally. SMC augments this with a resampling step to mitigate weight degeneracy. For $N$ particles, SMC proceeds as follows:

1. Initialise: $x_T^{(i)} \sim \pi(x_T), w_T^{(i)} = 1$.

2. For $t = T, \ldots, 1$:

    (a) Propagate: $x_{t-1}^{(i)} \sim q(x_{t-1}|x_t^{(i)})$.

    (b) Update weights: $\frac{\pi(x_t^{(i)})\gamma(x_{t+1}^{(i)}|x_t^{(i)})}{\pi(x_{t+1})q(x_t^{(i)}|x_{t+1}^{(i)})}$.

    (c) Resample particles according to $\left\{ \frac{w_{t-1}^{(i)}}{\sum_{j=1}^{N} w_{t-1}^{(j)}} \right\}_{i=1}^{N}$; then reset all weights to $w_{t-1}^{(i)} = 1$.

The resulting set of particles $\{x_0^{(i)}, w_0^{(i)}\}_{i=1}^{N}$ forms an empirical approximation of the target $\pi(x_0)$.

## A.2 PROOF OF LOCALLY OPTIMAL PROPOSAL

**Proposition 2** (Locally Optimal Proposal). *Given the incremental importance weight as in Equation* (5) $w_{t-1}(x_{t-1}, x_t) = \frac{\pi(x_{t-1})\gamma(x_t|x_{t-1})}{\pi(x_t)q(x_{t-1}|x_t)}$, *the proposal distribution that minimises the variance of* $w_{t-1}$, *often referred to as the* locally optimal proposal, *is* $q(x_{t-1}|x_t) \propto \pi(x_{t-1})\gamma(x_t|x_{t-1})$.

*Proof.* We first present an intuitive argument to aid understanding, and subsequently provide the formal proof.

*Intuitive argument.* The optimal proposal distribution is characterised as the one that minimises the variance of the importance weights. In the degenerate case of zero variance, the importance weight must be constant: $\frac{\pi(x_{t-1})\gamma(x_{t-1}|x_t)}{\pi(x_{t-1})q(x_{t-1}|x_t)} = c$, for some constant $c > 0$. Rearranging yields

$$
q^*(x_{t-1}|x_t) = \frac{1}{c} \frac{\pi(x_{t-1})}{\pi(x_t)} \gamma(x_{t-1}|x_t) \propto \pi(x_{t-1})\gamma(x_{t-1}|x_t),
\tag{14}
$$

where $c = \frac{1}{\pi(x_t)} \sum_{x_{t-1}} \pi(x_{t-1})\gamma(x_{t-1}|x_t)$ is the normalising constant.

*Formal proof.* The optimal proposal can be obtained by minimising the variance of the incremental importance weight $w(x_{t-1}, x_t) = \frac{\pi(x_{t-1})\gamma(x_{t-1}|x_t)}{\pi(x_{t-1})q(x_{t-1}|x_t)}$:

$$
\begin{aligned}
q^* &= \operatorname*{argmin}_{q} \mathbb{E}_q \left[ w(x_{t-1}, x_t) - \mathbb{E}_q[w(x_{t-1}, x_t)] \right]^2 + a \left( \sum_{x_{t-1}} q(x_{t-1}|x_t) - 1 \right) \\
&= \operatorname*{argmin}_{q} \mathbb{E}_q \left[ w(x_{t-1}, x_t)^2 \right] - \mathbb{E}_q[w(x_{t-1}, x_t)]^2 + a(x_t) \left( \sum_{x_{t-1}} q(x_{t-1}|x_t) - 1 \right), \\
&= \operatorname*{argmin}_{q} \sum_{x_{t-1}} \underbrace{w(x_{t-1}, x_t)^2 q(x_{t-1}|x_t) + a(x_t)q(x_{t-1}|x_t)}_{:=F(q)} + c,
\end{aligned}
$$

where $c$ denotes a constant w.r.t. $q$ and we introduce a Lagrange multiplier $a(x_t) > 0$ for the constraint $\sum_{x_{t-1}} q(x_{t-1}|x_t) = 1$. Using the calculation of variation, where the functional $F$ should satisfy the Euler-Lagrange equation $\frac{\partial F}{\partial q} - \frac{\mathrm{d}}{\mathrm{d}x} \frac{\partial F}{\partial q'} = 0$, we have

$$\frac{\partial F}{\partial q} = -\left(\frac{\pi(x_{t-1})\gamma(x_{t-1}|x_t)}{\pi(x_{t-1})q(x_{t-1}|x_t)}\right)^2 + a(x_t) = 0 \implies q^*(x_{t-1}|x_t) = \frac{\pi(x_{t-1})\gamma(x_{t-1}|x_t)}{\pi(x_t)\sqrt{a(x_t)}}.$$

The term $\frac{1}{\pi(x_t))\sqrt{a(x_t)}}$ is a normalisation constant that does not depend on the $x_{t-1}$. We can find its value by enforcing the constraint $\sum_{x_{t-1}} q^*(x_{t-1}|x_t) = 1$. This shows that the optimal proposal is $q^*(x_{t-1}|x_t) \propto \pi(x_{t-1})\gamma(x_{t-1}|x_t)$. $\qquad\square$

**Remark.** Given Proposition 2, we can derive the form of the locally optimal proposal under different settings. Specifically, using the importance weight defined in Equations (6) and (7), let the forward kernel $\gamma(x_t|x_{t-1})$ be specified as

$$\text{product: } \gamma \propto p_1^\alpha(x_t|x_{t-1})p_2^\beta(x_t|x_{t-1}) \quad \text{reward-tilting: } \gamma \propto p(x_{t-1}|x_t)$$

The corresponding importance weights are then

$$\text{product: } \frac{p_{\theta_1}^\alpha(x_{t-1}|x_t)p_{\theta_2}^\beta(x_{t-1}|x_t)}{Z(x_{t-1})q(x_{t-1}|x_t)} \quad \text{reward-tilting: } \frac{\exp(r(x_{t-1}))}{\exp(r(x_t))}\frac{p_\theta(x_{t-1}|x_t)}{q(x_{t-1}|x_t)}$$

where $Z(x_{t-1}) = \sum_{x_t} p_1^\alpha(x_t|x_{t-1})p_2^\beta(x_t|x_{t-1})$ is the normalising constant. By Proposition 2, the corresponding locally optimal proposals are

$$\text{product: } q \propto \frac{p_{\theta_1}^\alpha(x_{t-1}|x_t)p_{\theta_2}^\beta(x_{t-1}|x_t)}{Z(x_{t-1})} \quad \text{reward-tilting: } q \propto \exp(r(x_{t-1}))p_\theta(x_{t-1}|x_t)$$

The normalising constant $Z$ is tractable, since $p(x_t|x_{t-1})$ is a simple forward noising distribution (induced by Equation (1)) that does not involve network evaluation. In contrast, for the reward-tilting, the dependence on the reward function $r$, which is defined via a neural network, renders the optimal proposal intractable in general. This necessitates the development of approximation techniques.

## A.3 PROOF OF LOG-VARIANCE MINIMISATION OBJECTIVE

**Corollary 1.** *The locally optimal proposal $q^* \propto \pi(x_{t-1})p_\theta(x_{t-1}|x_t)$ that achieves the minimum variance of the important weight $\mathbb{V}_q\left[\frac{\pi(x_{t-1})\gamma(x_t|x_{t-1})}{\pi(x_t)q(x_{t-1}|x_t)}\right]$ is unique.*

*Proof.* Recall that the variance is given by $\mathbb{V}_q[w] = \mathbb{E}_q[w^2] - (\mathbb{E}_q[w])^2$. As shown in the proof of Proposition 2, the term $\mathbb{E}_q[w]$ is constant w.r.t. the choice of $q$. Therefore, minimising the variance is equivalent to minimise the expected square of the weights, $\mathbb{E}_q[w^2]$, which we will call $F(q)$:

$$F(q) = \mathbb{E}_q[w^2] = \sum_{x_{t-1}} \frac{1}{q(x_{t-1}|x_t)}\left(\frac{\pi(x_{t-1})\gamma(x_t|x_{t-1})}{\pi(x_t)}\right)^2. \tag{15}$$

To simplify the notation, let $q_i = q(x_{t-1} = i|x_t)$ and $C_i = \frac{\pi(x_{t-1}=i)\gamma(x_t|x_{t-1}=i)}{\pi(x_t)}$. The optimal proposal is $q^* = C_i/Z$, where $Z = \sum_j C_j$. We then can rewrite the objective function $F(q)$ as

$$F(q) = \sum_i \frac{C_i^2}{q_i} = \sum_i \frac{(Zq_i^*)^2}{q_i} = Z^2 \sum_i \frac{(q_i^*)^2}{q_i}. \tag{16}$$

Evaluating the function at the optimum, $q^*$, we have

$$F(q^*) = \sum_i \frac{C_i^2}{q_i^*} = \sum_i \frac{(Zq_i^*)^2}{q_i^*} = Z^2 \sum_i q_i^* = Z^2. \tag{17}$$

To prove the uniqueness of the locally optimal proposal $q^*$, the key insight is to relate the expression for $F(q)$ to the Chi-squared divergence, which is defined as

$$\chi^2(q^*\|q) = \sum_i \frac{(q_i^* - q_i)^2}{q_i} = \left(\sum_i \frac{(q_i^*)^2}{q_i}\right) - 1. \tag{18}$$

Rearranging this, we see that $\sum_i \frac{(q_i^*)^2}{q_i} = \chi^2(q^*\|q) + 1$. Now we can express $F(q)$ as

$$F(q) = Z^2 \sum_i \frac{(q_i^*)^2}{q_i} = Z^2 \left(\chi^2(q^*\|q) + 1\right) = Z^2\chi^2(q^*\|q) + Z^2. \tag{19}$$

Since $F(q^*) = Z^2$, we finally arrive at

$$F(q) = F(q^*) + Z^2\chi^2(q^*\|q). \tag{20}$$

Since the $\chi^2$-divergence is non-negative and $\chi^2(q^*\|q) = 0$ if and only if $q = q^*$, we see that the equality $F(q) = F(q^*)$ holds only when $\chi^2(q^*\|q) = 0$, which requires that $q$ be identical to $q^*$. For any other distribution $q \neq q^*$, the divergence is strictly positive, meaning $F(q) > F(q^*)$. Therefore, $q^*$ is the unique distribution that minimises the variance of the importance weights. $\square$

**Proposition 3.** *For any reference distribution $q_{\text{ref}}$, we have $\mathcal{L}_{\text{log-var}}(\phi) \leq T^2\mathcal{L}(\phi, \psi)$. Moreover, the minimiser of $\mathcal{L}$ is unique and attains its optimum when $q_\phi \propto \exp(r(x_{t-1}))p_\theta(x_{t-1}|x_t)$.*

*Proof.* To prove the result, we first recall the basic identitie $\mathbb{E}_q[w] = \operatorname{argmin}_c \mathbb{E}_q[(w-c)^2]$ and $\mathbb{V}_q[w] = \mathbb{E}_q[(w - \mathbb{E}_q[w])^2]$. Applying these, we obtain

$$\mathcal{L}_{\text{log-var}}(\phi) = \mathbb{V}_{q_{\text{ref}}}\left[\sum_t \log \frac{\exp(r(x_{t-1}))\, p_\theta(x_{t-1}|x_t)}{\exp(r(x_t))\, q_\phi(x_{t-1}|x_t)}\right]$$

$$= \min_{F_t \in \mathbb{R}} \mathbb{E}_{q_{\text{ref}}}\left[\left|\sum_t \log \frac{\exp(r(x_{t-1}))\, p_\theta(x_{t-1}|x_t)}{\exp(r(x_t))\, q_\phi(x_{t-1}|x_t)} - F_t\right|^2\right]$$

$$= T^2 \min_{F_t \in \mathbb{R}} \mathbb{E}_{q_{\text{ref}}}\left[\left|\sum_t \frac{1}{T} \log \frac{\exp(r(x_{t-1}))\, p_\theta(x_{t-1}|x_t)}{\exp(r(x_t))\, q_\phi(x_{t-1}|x_t)} - \frac{1}{T}F_t\right|^2\right]$$

$$\leq T^2 \min_{F_t \in \mathbb{R}} \mathbb{E}_{q_{\text{ref}}}\left[\sum_t \frac{1}{T}\left|\log \frac{\exp(r(x_{t-1}))\, p_\theta(x_{t-1}|x_t)}{\exp(r(x_t))\, q_\phi(x_{t-1}|x_t)} - F_t\right|^2\right]$$

$$= T^2 \min_{F_t \in \mathbb{R}} \mathbb{E}_{q_{\text{ref}},t}\left[\left|\log \frac{\exp(r(x_{t-1}))\, p_\theta(x_{t-1}|x_t)}{\exp(r(x_t))\, q_\phi(x_{t-1}|x_t)} - F_t\right|^2\right].$$

This motivates defining the loss function as

$$\mathcal{L}(\theta, \psi) = \mathbb{E}_{t, x_t \sim q_{\text{ref}}}\left[\left|\log \frac{\exp(r(x_{t-1}))\, p_\theta(x_{t-1}|x_t)}{\exp(r(x_t))\, q_\phi(x_{t-1}|x_t)} - F_\psi(t)\right|^2\right]. \tag{21}$$

Therefore, we have the inequality

$$\mathcal{L}_{\text{log-var}}(\phi) \leq T^2\mathcal{L}(\theta, \psi) \tag{22}$$

Consequently, if $(\theta^*, \psi^*) = \operatorname{argmin}_{\phi, \psi} \mathcal{L}(\theta, \psi)$, then

$$\mathcal{L}(\theta^*, \psi^*) = 0 \quad \Rightarrow \quad \mathcal{L}_{\text{log-var}}(\phi^*) = 0 \quad \Rightarrow \quad \mathbb{V}[w] = 0. \tag{23}$$

Finally, by Corollary 1, the minimiser $\phi^*$ is unique, and the corresponding proposal takes the form $q_{\phi^*}(x_{t-1}|x_t) \propto \exp(r(x_{t-1}))p_\theta(x_{t-1}|x_t)$. $\square$

## B EXTENDING DISCRETE-TIME SMC TO CONTINUOUS-TIME SMC

In this section, we extend our SMC algorithm from discrete time to continuous time. We begin by introducing the key preliminary: the continuous-time Markov chain (CTMC) (Norris, 1998). We then establish connections to previous work Holderrieth et al. (2025); Lee et al. (2025), which develops continuous-time SMC methods for discrete diffusion models.

## B.1 Background of CTMC

A continuous-time Markov chain (Norris, 1998) at time $t$ is characterised by a time-dependent rate matrix $R_t : \mathcal{X} \times \mathcal{X} \to \mathbb{R}$, which captures the instantaneous rate of change of the transition probabilities. Specifically, the rate matrix $R_t$ is defined as

$$R_t(x, y) = \lim_{\Delta t \to 0} \frac{p_{t+\Delta t | t}(y|x) - \delta_{y=x}}{\Delta t}, \quad \delta_{y=x} = \begin{cases} 1, & y = x \\ 0, & y \neq x \end{cases}. \tag{24}$$

By definition, the rate matrix equivalently yields the transition probability

$$p_{t+\Delta t | t}(y|x) = \delta_{y=x} + R_t(x, y)\Delta t + \mathcal{O}(\Delta t). \tag{25}$$

To ensure $p_{t+\Delta t|t}$ be a valid distribution, the rate matrix $R_t$ must satisfy the following constraints:

$$R_t(x, y) \geq 0, \forall y \neq x, \quad R_t(x, x) = -\sum_{y \neq x} R_t(x, y). \tag{26}$$

The transition probability $p_{t|s}$, for $t > s$, satisfies the Kolmogrove equations (Oksendal, 2013):

$$\text{Kolmogorov forward equation:} \quad \partial_t p_{t|s}(x|\tilde{x}) = \sum_y p_{t|s}(y|\tilde{x}) R_t(y, x) \tag{27}$$

$$\text{Kolmogorov backward equation:} \quad \partial_s p_{t|s}(x|\tilde{x}) = -\sum_y R_t(\tilde{x}, y) p_{t|s}(x|y) \tag{28}$$

The forward equation also induces a PDE for the marginal distribution $p_t(x)$

$$\partial_t p_t(x) = \sum_y p_t(y) R_t(y, x). \tag{29}$$

Using the backward equation, one can derive a Kolmogorov backward equation for expectations, also known as Dynkin's formula (Oksendal, 2013). In particular, we have the following lemma.

**Lemma 1.** *Let $h$ be a test function of interest and define $u_t(x) = \mathbb{E}_{p_{1|t}(z|x)}[h(z)]$. Then $u_t$ satisfies the partial differential equation $\partial_t u_t(x) = -\sum_y R_t(x, y) u_t(y)$.*

*Proof.*

$$\partial_t u_t(x) = \sum_z h(z) \partial_t p_{1|t}(z|x)$$

$$= \sum_z h(z) - \sum_y R_t(x, y) p_{1|t}(z, y)$$

$$= -\sum_y R_t(x, y) \sum_z p_{1|t}(z, y) h(z)$$

$$= -\sum_y R_t(x, y) u_t(y).$$

$\square$

## B.2 Continuous-Time Formulation of SMC

**Proposition 1** (SMC for Continuous-Time Discrete Diffusion). *Let $R_t$ be the rate matrix generating the forward transition kernel $\gamma(x_t|x_{t-\Delta t})$, and $\hat{R}_t$ be its counterpart associated with the backward proposal kernel $q(x_{t-\Delta t}|x_t)$, where $\Delta t \to 0$ is the infinitesimal time increment. Then, the importance weight at time $t$ is given by $w_t = \int_1^t -\partial_s \log \pi(x_s) + \sum_{y_s} R_s(x_s, y_s) \frac{\pi(y_s)}{\pi(x_s)} \, ds$, if the forward kernel $\gamma$ is chosen such that the rate matrices satisfy detailed balance $\hat{R}_t(x_t, y_t)\pi(x_t) = R_t(y_t, x_t)\pi(y_t)$.*

*Proof.* Recall Equation (5), where the importance weight is given by

$$w_{t-1}(x_{t-1:T}) = \frac{\pi(x_{t-1})}{\pi(x_t)} \frac{\gamma(x_t|x_{t-1})}{q(x_{t-1}|x_t)} w_t(x_{t:T}). \tag{30}$$

We now extend it to the continuous-time setting. Let $R_t$ and $\hat{R}_t$ denote the rate matrices corresponding to the proposal $q$ and the forward noising transition $\gamma$, respectively. Consider a discretisation with $T$ denoising steps, indexed by time points $s = t_0 < \cdots < t_i \cdots < t_T = 1$, where each interval satisfies $t_i - t_{i-1} = \frac{1-s}{T}$. The discrete-time importance weight at step time $s$ is then computed as

$$\log w_s = \log \frac{\pi(x_s)}{\pi(x_1)} + \sum_{i=1}^{T} \log \frac{\gamma(x_{t_i}|x_{t_{i-1}})}{q(x_{t_{i-1}}|x_{t_i})}. \tag{31}$$

The second term in the RHS can be expanded as

$$\sum_i \log \frac{\gamma(x_{t_i}|x_{t_{i-1}})}{q(x_{t_{i-1}}|x_{t_i})} = \sum_i \log\left(\delta_{x_{t_i}=x_{t_{i-1}}} + \hat{R}_{t_i}(x_{t_{i-1}}, x_{t_i})\frac{1}{T}\right) - \log\left(\delta_{x_{t_{i-1}}=x_{t_i}} + R_{t_i}(x_{t_i}, x_{t_{i-1}})\frac{1}{T}\right)$$

$$= \sum_{i,t_i=t_{i-1}} \log\left(1 + \hat{R}_{t_i}(x_{t_i}, x_{t_i})\frac{1}{T}\right) - \log\left(1 + R_{t_i}(x_{t_i}, x_{t_i})\frac{1}{T}\right) + \sum_{i,t_i \neq t_{i-1}} \hat{R}_{t_i}(x_{t_{i-1}}, x_{t_i}) - R_{t_i}(x_{t_i}, x_{t_{i-1}})$$

$$= \sum_{i,t_i=t_{i-1}} \hat{R}_{t_i}(x_{t_i}, x_{t_i})\frac{1}{T} - R_{t_i}(x_{t_i}, x_{t_i})\frac{1}{T} + \mathcal{O}(\frac{1}{T}) + \sum_{i,t_i \neq t_{i-1}} \hat{R}_{t_i}(x_{t_{i-1}}, x_{t_i}) - R_{t_i}(x_{t_i}, \mathbf{x}_{t_{i-1}})$$

Taking the limit $T \to +\infty$, the importance weight becomes:

$$\log w_s = \log \frac{\pi(x_s)}{\pi(x_1)} + \int_1^s R_t(x_t, x_t) - \hat{R}_t(x_t, x_t)\, dt + \sum_{s \leq t, x_{t+} \neq x_t} \log \hat{R}_t(x_t, x_{t+}) - \log R_t(x_{t+}, x_t). \tag{32}$$

By the fundamental theorem of calculus for piecewise differentiable functions, we have:

$$\log \frac{\pi(x_s)}{\pi(x_1)} = \int_1^s -\partial_t \log \pi(x_t) + \sum_{s \leq t, x_{t+} \neq x_t} \log \pi(x_t) - \log \pi(x_{t+}). \tag{33}$$

If the noising process $\gamma$ is chosen such that the rate matrix satisfies $\hat{R}_t(x_t, y_t)\pi(x_t) = R_t(y_t, x_t)\pi(y_t)$, then the importance weight simplifies accordingly

$$\log w_s = \int_1^s -\partial_t \log \pi(x_t) + R_t(x_t, x_t) - \hat{R}_t(x_t, x_t)\, dt$$

$$= \int_1^s -\partial_t \log \pi(x_t) + \sum_{y_t} R_t(x_t, y_t)\frac{\pi(y_t)}{\pi(x_t)}\, dt,$$

which completes the proof. $\qquad\square$

**Remark.** Proposition 1 recovers the importance weights proposed in the SMC methods of Holderrieth et al. (2025); Ou et al. (2025). In those works, the intermediate target distribution is defined as a geometric interpolation between the base and target distributions: $\pi(x_t) \propto p_{\text{base}}^t(x_t)p_{\text{target}}^{1-t}(x_t)$. The proposal rate matrix $R_t$ is then trained to satisfy the Kolmogorov forward equation. In contrast, we consider a different scenario: the pretrained model is available, but the intermediate target $\pi(x_t)$ cannot be computed explicitly. Moreover, the importance weight in Equation (32) can also be derived via the Radon–Nikodym derivative (Campbell et al., 2024, Appendix C.1), (Denker et al., 2025, Lemma 4). We instead adopt a discrete-time formulation and present a streamlined derivation to keep the exposition accessible for readers who may not be familiar with Radon–Nikodym derivative or path-measure theory.

We next extend the discrete SMC framework to the continuous-time setting, concentrating on the reward-tilting formulation. While this formulation has also been considered in Lee et al. (2025), our treatment proceeds from a distinct perspective. Before proceeding with the main development, we establish several auxiliary lemmas that are essential for the subsequent derivations.

**Lemma 2.** *For a continuous time Markov chain with distribution $p$ and rate matrix $R$, the rate matrix for the reverse process satisfy $\hat{R}_t(x_t, y_t) = R_t(y_t, x_t)\frac{p(y_t)}{p(x_t)}$ and $\hat{R}_t(x_t, x_t) = -\sum_{y_t \neq x_t} \hat{R}_t(x_t, y_t) = -\sum_{y_t \neq x_t} R_t(y_t, x_t)\frac{p(y_t)}{p(x_t)}$.*

*Proof.* See (Sun et al., 2022, Appendix B.2) for a detailed proof. $\square$

**Lemma 3.** *For a continuous time Markov chain with distribution $p$ and rate matrix $R$, it satisfies*

$$\partial_t \log p(x_t) = \sum_{y_t \neq x_t} R_t(y_t, x_t)\frac{p(y_t)}{p(x_t)} + R_t(x_t, x_t). \tag{34}$$

*Proof.* By applying the forward Kolmogrov equation in Equation (29), we have

$$\partial_t \log p(x_t) = \frac{\partial_t p(x_t)}{p(x_t)} = \frac{1}{p(x_t)}\sum_{y_t} R_t(y_t, x_t)p(y_t) = \sum_{y_t \neq x_t} R_t(y_t, x_t)\frac{p(y_t)}{p(x_t)} + R_t(x_t, x_t) \tag{35}$$

which completes the proof. $\square$

**Lemma 4.** *For a continuous time Markov chain with distribution $p$ and rate matrix $R$, the function $u(x_t) = \mathbb{E}_{p(x_0|x_t)}[\exp(r(x_0))]$ satisfies*

$$\partial_t \log u(x_t) = R_t^{\alpha=1}(x_t, x_t) - R_t(x_t, x_t), \tag{36}$$

*where $R_t^{\alpha=1}(x_t, y_t) = R_t(x_t, y_t)\frac{u(y_t)}{u(x_t)}$ and $R_t^{\alpha=1}(x_t, x_t) = -\sum_{y_t \neq x_t} R_t^{\alpha=1}(x_t, y_t)$.*

*Proof.* By applying Lemma 1, we have

$$\partial_t \log u(x_t) = \frac{\partial_t u(x_t)}{u(x_t)} = -\frac{1}{u(x_t)}\sum_{y_t} R_t(x_t, y_t)u(y_t)$$

$$= -\sum_{y_t \neq x_t} R_t(x_t, y_t)\frac{u(y_t)}{u(x_t)} - R_t(x_t, x_t)$$

$$= R_t^{\alpha=1}(x_t, x_t) - R_t(x_t, x_t),$$

which completes the proof. $\square$

We are now ready to prove the result in Lee et al. (2025).

**Proposition 4** (Continuous-Time SMC for Reward-Tilting (Lee et al., 2025))**.** *Let $p_\theta(x_t)$ denote a pretrained diffusion model, $R_t$ the rate matrix generating the desnoising probability path, and $\hat{R}_t$ the corresponding rate matrix for the forward noising path. The intemediate target distributino is defined as $\pi(x_t) = p_\theta(x_t)u^\alpha(x_t)$, where $u(x_t) = \mathbb{E}_{p_\theta(x_0|x_t)}[\exp(r(x_0))]$ is the reward-tilting functioin. Let $Q_t$ be the proposal rate matrix in SMC; the importance weight is then given by*

$$\log w_s = \int_1^s Q_t(x_t, x_t) - R_t(x_t, x_t)\,\mathrm{d}t + \sum_{s \leq t, x_{t+} \neq x_t} \log R_t(x_{t+}, x_t) - \log Q_t(x_{t+}, x_t)$$

$$+ \int_1^s \alpha\left(R_t(x_t, x_t) - R_t^{\alpha=1}(x_t, x_t)\right)\mathrm{d}t + \sum_{s \leq t, x_{t+} \neq x_t} \alpha\left(\log R_t^{\alpha=1}(x_{t+}, x_t) - \log R_t(x_{t+}, x_t)\right),$$

*where $R_t^{\alpha=1}(x_t, y_t) = R_t(x_t, y_t)\frac{u(y_t)}{u(x_y)}$ and $R_t^{\alpha=1}(x_t, x_t) = -\sum_{y_t \neq x_t} R_t(x_t, y_t)\frac{u(y_t)}{u(x_y)}$.*

*Proof.* By the derivation of Proposition 1, it gives that the importance weight takes the form

$$\log w_s = \underbrace{\int_1^s -\partial_t \log \pi(x_t) + Q_t(x_t, x_t) - \hat{R}_t(x_t, x_t)\,\mathrm{d}t}_{①}$$

$$+ \underbrace{\sum_{s \leq t, x_{t+} \neq x_t} \log \pi(x_t) - \log \pi(x_{t+}) + \log \hat{R}_t(x_t, x_{t+}) - \log Q_t(x_{t+}, x_t)}_{②}. \tag{37}$$

By applying Lemmas 2 to 4, we can expand ① as

$$\text{①} = \int_1^s -\left(\alpha\partial_t \log u(x_t) + \sum_{y_t \neq x_t} R_t(y_t, x_t)\frac{p_\theta(y_t)}{p_\theta(x_t)} + R_t(x_t, x_t)\right) + Q_t(x_t, x_t) + \sum_{y_t \neq x_t} R_t(y_t, x_t)\frac{p_\theta(y_t)}{p_\theta(x_t)} \, \mathrm{d}t$$

$$= \int_1^s -\alpha\partial_t \log u(x_t) - R_t(x_t, x_t) + Q_t(x_t, x_t) \, \mathrm{d}t$$

$$= \int_1^s \alpha\left(R_t(x_t, x_t) - R_t^{\alpha=1}(x_t, x_t)\right) - R_t(x_t, x_t) + Q_t(x_t, x_t) \, \mathrm{d}t.$$

Similarly, by applying Lemma 2, ② follows

$$\text{②} = \sum_{s \leq t, x_{t^+} \neq x_t} \alpha\left(\log u(x_t) - \log u(x_{t^+})\right) + \log R_t(x_{t^+}, x_t) - \log Q_t(x_{t^+}, x_t)$$

$$= \sum_{s \leq t, x_{t^+} \neq x_t} \alpha\left(\log R_t^{\alpha=1}(x_{t^+}, x_t) - \log R_t(x_{t^+}, x_t)\right) + \log R_t(x_{t^+}, x_t) - \log Q_t(x_{t^+}, x_t).$$

where the second equation follows from the identity

$$\log u(y_t) - \log u(x_t) = \log R_t^{\alpha=1}(x_t, y_t) - \log R_t(x_t, y_t), \tag{38}$$

which is followed by the definition of $R_t(x_t, y_t)$. Combining ① and ②, the full expression for the importance weight becomes

$$\log w_s = \int_1^s Q_t(x_t, x_t) - R_t(x_t, x_t) \, \mathrm{d}t + \sum_{s \leq t, x_{t^+} \neq x_t} \log R_t(x_{t^+}, x_t) - \log Q_t(x_{t^+}, x_t)$$

$$+ \int_1^s \alpha\left(R_t(x_t, x_t) - R_t^{\alpha=1}(x_t, x_t)\right) \mathrm{d}t + \sum_{s \leq t, x_{t^+} \neq x_t} \alpha\left(\log R_t^{\alpha=1}(x_{t^+}, x_t) - \log R_t(x_{t^+}, x_t)\right),$$

which completes the proof. □

## C  IMPLEMENTATION DETAILS OF COMPUTING IMPORTANCE WEIGHT

In masked diffusion models, although ancestor sampling (Austin et al., 2021; Sahoo et al., 2024; Shi et al., 2024) is the de facto method for inference, low-confidence sampling (Chang et al., 2022) is more widely used in practice due to its stronger empirical performance. However, this approach makes it challenging to explicitly compute the importance weights. In this section, we first provide a brief recap of the main sampling schemes used in masked diffusion models, and then present a method to address the difficulty of computing importance weights under low-confidence sampling.

### C.1  SAMPLING SCHEMES IN MASKED DIFFUSION MODELS

**MDM Sampling (Sahoo et al., 2024).** MDM sampling is the de facto method for inference in masked diffusion models. Given a trained denoiser $\mu_\theta$, which predicts the clean data $x_0$, MDM sampling performs ancestor sampling to generate samples according to

$$p_\theta(x_{t-1}|x_t) = \begin{cases} \text{Cat}(x_{t-1}; x_t) & x_t \neq [\text{m}] \\ \text{Cat}\left(x_{t-1}; \frac{(1-\alpha_{t-1})[\text{m}] + (\alpha_{t-1} - \alpha_t)\mu_\theta(x_t)}{1-\alpha_t}\right) & x_t = [\text{m}] \end{cases} \tag{39}$$

While theoretically sound, a major limitation of MDM sampling is that once a latent variable $x_t$ is assigned a non-mask category during the unmasking process, it becomes immutable. Consequently, any errors made during unmasking are irreversible and persist in the final generated samples.

**ReMDM Sampling (Wang et al., 2025).** ReMDM sampling is a modification of MDM that allows previously unmasked tokens to be remasked during the unmasking process. The posterior is constructed so that the forward marginal $p(x_t|x_0)$ remains identical to that of masked diffusion in Equation (1):

$$p_\sigma(x_{t-1}|x_t, x_0) = \begin{cases} \text{Cat}(x_{t-1}; (1-\sigma_t)x_t + \sigma_t[\text{m}]) & x_t \neq [\text{m}] \\ \text{Cat}\left(x_{t-1}; \frac{\alpha_{t-1} - (1-\sigma_t)\alpha_t}{1-\alpha_t}x_0 + \frac{1-\alpha_{t-1} - \sigma_t\alpha_t}{1-\alpha_t}[\text{m}]\right) & x_t = [\text{m}] \end{cases}. \tag{40}$$

Here $\sigma_t$ is the remasking schedule. To ensure the posterior remains valid, it must satisfy the constraint:

$$0 \leq \sigma_t \leq \min\left\{1, \frac{1 - \alpha_{t-1}}{\alpha_t}\right\}. \tag{41}$$

The reverse unmasking process is then parameterised as

$$p_\theta(x_{t-1}|x_t) = p_\sigma(x_{t-1}|x_t, \mu_\theta(x_t)). \tag{42}$$

Notably, the ReMDM training objective is a reweighted version of the standard masked diffusion loss in Equation (3). Thus, we can take a pretrained masked diffusion model, and use the ReMDM sampling in Equation (42) for inference.

**Low-Confidence Sampling (Chang et al., 2022).** Low-confidence sampling is the most commonly used method in discrete diffusion. In brief, at each denoising step, the denoiser $\mu_\theta$ predicts the clean data $x_0$, and tokens with low confidence, which are measured as the maximum logit of $\mu_\theta$ at each position, are selectively remasked for further refinement. Formally, the reverse unmasking process can be parametrised as

$$p_\theta(x_{t-1}|x_t) = \sum_{x_0} p_\theta(x_0|x_t)\mathbf{1}_{x_{t-1}[l]=x_0[l]}, \quad l = \underset{l \in \{1,...,L\}}{\text{argmax}} \max(\mu_\theta(x_t)[l]) \wedge x_t[l] = [\text{m}], \tag{43}$$

where $x[l]$ denotes the $l$-th token of $x$ of $L$ length, and $\max(v)$ returns the maximum value of the vector $v$. For clarity, here we only consider unmasking a single token at each step. In practice, multiple tokens can be unmasked simultaneously by using the same strategy.

## C.2 FIRST-ORDER APPROXIMATION WITH GUMBEL-SOFTMAX RELAXATION

To apply the first-order approximately optimal proposal, we need to compute $\nabla_{x_t}\hat{r}(x_t)$, where $\hat{r}(x_t) = \frac{1}{M}\sum_{m=1}^{M} r(x_0^{(m)})$, $x_0^{(m)} \sim p_\theta(x_0|x_t)$ as defined in Equation (12). However, because $x_0^{(m)}$ is drawn via categorical sampling, $\hat{r}(x_t)$ is not differentiable with respect to $x_t$. To address this, we use the Gumbel–Softmax reparameterization trick to obtain a differentiable surrogate. Concretely, we break the computation of $\hat{r}(x_t)$ into three steps:

1. compute the denoising logits: $p = \mu_\theta(x_t)$, where $\mu_\theta$ is the denoising model;

2. sample $x_0$: $x_0^{(m)} \sim \text{Cat}(x; p)$;

3. evaluate the reward: $\hat{r}(x_t) = \frac{1}{M}\sum_{m=1}^{M} r(x_0^{(m)})$.

Following Grathwohl et al. (2021), we treat both $r$ and $\mu_\theta$ as functions that accept continuous inputs so that their gradients are well defined (steps 1 and 3). For step 2, we replace the categorical draw with its Gumbel–Softmax relaxation (Jang et al., 2016), making the sample $x_0^{(m)}$ differentiable with respect to $p$. Using these relaxations, the gradient can be obtained by the chain rule:

$$\nabla_{x_t}\hat{r}(x_t) \approx \frac{1}{M}\sum_{m=1}^{M} \frac{\partial r(x_0^{(m)})}{\partial x_0^{(m)}} \frac{\partial x_0^{(m)}}{\partial p} \frac{\partial p}{\partial x_t}, \quad x_0^{(m)} \sim p_\theta(x_0|x_t). \tag{44}$$

The first and last factors are provided by the differentiability of $r$ and $\mu_\theta$, while the middle term is approximated using the Gumbel–Softmax relaxation.

**Remark.** The gradient approximation in Equation (44) requires the reward $r(x_0)$ to be differentiable w.r.t. $x_0$. In the setting of non-differentiable rewards, one can apply the REINFORNCE gradient estimator (Williams, 1992). Specifically, we have

$$\nabla_{x_t}\hat{r}(x_t) = \nabla_{x_t}\mathbb{E}_{p_\theta(x_0|x_t)}[r(x_0)]$$

$$= \sum_{x_0} p_\theta(x_0|x_t)\nabla_{x_t}\log p_\theta(x_0|x_t)r(x_0)$$

$$= \mathbb{E}_{p_\theta(x_0|x_t)}[\nabla_{x_t}\log p_\theta(x_0|x_t)r(x_0)].$$

Thus, the gradient can be approximated via

$$\nabla_{x_t}\hat{r}(x_t) \approx \frac{1}{M}\sum_{m=1}^{M} r(x_0^{(m)})\nabla_{x_t}\log p_\theta(x_0^{(m)}|x_t), \quad x_0^{(m)} \sim p_\theta(x_0|x_t). \tag{45}$$

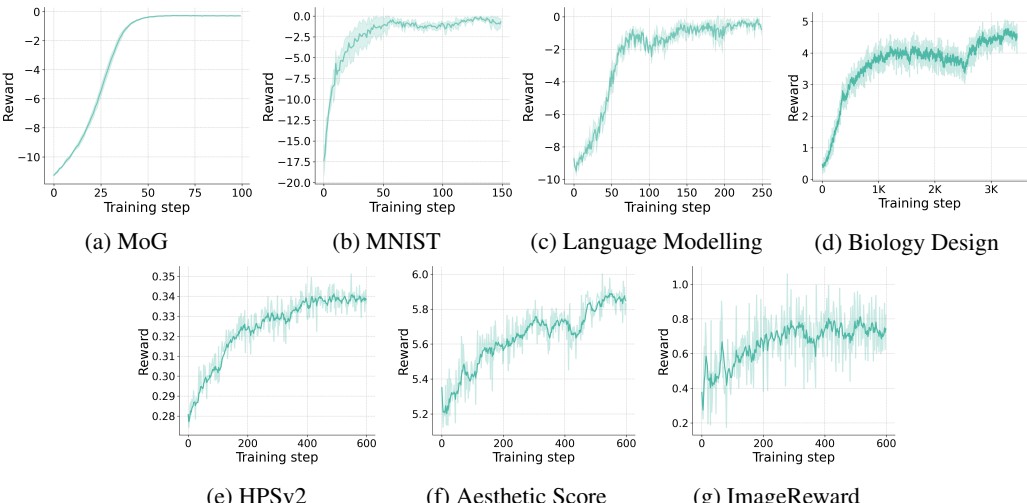

Figure 7: Reward convergence curves for different experiments throughout the finetuning process.

### C.3 Computing Importance Weight with Low-Confidence Sampling

To apply SMC to masked diffusion models, one must compute the log-ratio in the importance weight, as in Equation (7):

$$\log p_\theta(x_{t-1}|x_t) - \log q_\phi(x_{t-1}|x_t).$$ (46)

While this computation is straightforward for MDM and ReMDM sampling, it becomes tricky for low-confidence sampling. The difficulty arises because $p$ and $q$ rely on different denoisers, denoted as $\mu_\theta$ and $\mu_\phi$, respectively. If one strictly follows the rule in Equation (43), the log-ratio often collapses to zero whenever

$$l^p \neq l^q, \quad \text{where } l^q = \underset{l}{\arg\max} \max(\mu_\phi(x_t)[l]) \wedge x_t[l] = [\text{m}].$$ (47)

As a result, both SMC and the training objective in Equation (10) become ineffective in practice. To address this issue, we adopt a strategy in which both $p_\theta$ and $q_\phi$ use the same logit $\mu_\phi(x_t)$ to determine the remasked position $l$:

$$\begin{aligned} p_\theta(x_{t-1}|x_t) &= \sum_{x_0} p_\theta(x_0|x_t)\mathbf{1}_{x_{t-1}[l]=x_0[l]} \\ q_\phi(x_{t-1}|x_t) &= \sum_{x_0} q_\phi(x_0|x_t)\mathbf{1}_{x_{t-1}[l]=x_0[l]} \end{aligned} \qquad l = \underset{l\in\{1,...,L\}}{\arg\max} \max(\mu_\phi(x_t)[l]) \wedge x_t[l] = [\text{m}], \quad (48)$$

Under this formulation, the log-ratio can be computed as

$$\log\frac{p_\theta(x_{t-1}|x_t)}{q_\phi(x_{t-1}|x_t)} = \sum_{l\in\{1,...,L|x_{t-1}[l]\neq[\text{m}],x_t[l]=[\text{m}]\}} \log p_\theta(x_{t-1}[l] \mid x_t[l]) - \log q_\phi(x_{t-1}[l] \mid x_t[l])$$

(49)

This modification ensures that the importance weights remain well-defined under low-confidence sampling, enabling SMC to be applied effectively.

## D  Experimental Setting and Additional Results

In this section, we provide the details of experimental settings and addtional exeperiemental resuts.

### D.1  Details of Experimental Setting

We first describe the hyperparameters used in the SMC variants, and then discuss the training details of the learnable amortized proposal.

#### D.1.1  Choice of Hyperparameters in SMC

As described in Section 3.3, there are four key hyperparameters in our proposed SMC framework for the reward-tilting target: (i) the KL-regularization coefficient $\alpha$, (ii) the reward-twisted schedule $\lambda_t$, (iii) the number of Monte Carlo samples $M$, and (iv) the number of denoising steps $T$. Table 3 summarises

Table 3: Hyperparameters used in the SMC methods.

| | $\alpha$ | $\lambda_t$ | $M$ | $T$ |
|---|---|---|---|---|
| MoG | 1 | $1 - \frac{t}{T}$ | 10 | 100 |
| MNIST | 1 | $\min(1.05^{T-t} - 1, 1)$ | 10 | 100 |
| Language Modelling | 0.2 | $1 - \frac{t}{T}$ | 4 | 100 |
| Biology Design | 0.1 | $1 - \frac{t}{T}$ | 4 | 128 |
| Text-to-Image Generation | 0.01 | $1 - \frac{t}{T}$ | 1 | 48 |

the values of these hyperparameters used in our experiments. In practice, instead of using the mean to estimate the intermediate reward in Equation (12), we employ the log-sum-exp operation for improved stability, following Singhal et al. (2025):

$$\hat{r}(x_t) = \log\left(\frac{1}{M}\sum_{m=1}^{M}\exp(r(x_0^{(m)}))\right), \quad x_0^{(m)} \sim p_\theta(x_0|x_t). \tag{50}$$

Additionally, we provide ablation studies in Appendix D.2.1 to investigate the effects of $\lambda_t$ and $M$.

### D.1.2 TRAINING DETAILS OF THE AMORTISED PROPOSAL

**Synthetic Experiments.** In this experiment, we take the MDLM (Sahoo et al., 2024) as the pretrained diffusion model. Finetuning is performed on a single NVIDIA A6000 GPU with a batch size of 32 for MNIST and 128 for MoG. The model is trained for 30 epochs on MNIST and 20 epochs on MoG, with 5 optimisation steps per epoch. To avoid out-of-memory issues, we compute the loss over 10 randomly sampled time steps $t$ instead of using gradient accumulation, and choose $M = 10$ to estimate the reward. The Adam optimiser (Adam et al., 2014) is applied to train both the model and $F_\psi$, with a learning rate of 0.001 for MoG and 0.0001 for MNIST.

**Language Modelling.** This experiment closely follows Singhal et al. (2025). The pretrained language model used is MDLM[1] (Sahoo et al., 2024), which is trained on the OpenWebText dataset. We perform full-parameter finetuning on a single NVIDIA A6000 GPU with a batch size of 32. Training is conducted for 50 epochs, with 5 optimisation steps per epoch. To avoid the memory issue, at each optimisation step we compute the loss using one randomly selected time step $t$, together with a fixed $t = 0$. During training, rewards are scaled by a factor of 20, and estimated with $M = 20$ Monte Carlo samples. Both the model and $F_\psi$ are optimised using Adam (Adam et al., 2014) with a learning rate of 0.0001.

**Biology Design.** This experiment focuses on regulatory DNA sequence generation. We use the pretrained masked discrete diffusion model from Wang et al. (2024) which has been trained on a dataset of $\sim 700k$ DNA sequences (Gosai et al., 2023). We perform full-parameter finetuning on a single NVIDIA RTX 3090 GPU. Training is conducted for 350 epochs, with 10 optimisation steps per epoch. We use a batch size of 64 and use a sampling mix of $0.9 : 0.1$ of on-policy from $q_\phi$ and off-policy samples from $p_\theta$. To manage memory usage, we do a gradient accumulation of the loss at each timestep before taking an optimisation step. Additionally, we only consider the final 50 of the 128 timesteps for loss calculation, following Wang et al. (2024). We also add a negative entropy term of the form $\sum_{x_{t-1}} q_\phi(x_{t-1}|x_t)\log q_\phi(x_{t-1}|x_t)$ to the loss with a coefficient of 2.5[2]. We observe empirically that it helps in preventing mode collapse during training. For rewards, we use a scaling factor of 1000 and estimate it using just $M = 1$ Monte Carlo sample. Both the model and $F_\psi$ are optimized using the AdamW optimizer (Loshchilov & Hutter, 2017) with a learning rate of $1 \times 10^{-5}$.

**Text-to-Image Generation.** To finetune the Meissonic[3] model (Bai et al., 2024), we adopt low-rank adaptation (LoRA) (Hu et al., 2022) for parameter-efficient training. For training hyperparameters, we largely follow the DDPO (Black et al., 2023) implementation[4], with details provided here for completeness. All experiments are run on $8\times$NVIDIA H100 GPUs with a per-GPU batch size of 8. With 4-step gradient accumulation, this yields an effective batch size of 256. We train for 300 epochs, where each epoch consists of sampling 512 trajectories from the reference distribution $q_{\text{ref}}$ and performing 4 optimisation steps. The learning rate is fixed at $3 \times 10^{-4}$ for both the diffusion

---

[1] https://huggingface.co/kuleshov-group/mdlm-owt

[2] We empirically observe that the best value for the entropy coefficient varies proportionally with the reward scaling factor maintaining a ratio of $0.002 - 0.003$.

[3] https://huggingface.co/MeissonFlow/Meissonic

[4] https://github.com/kvablack/ddpo-pytorch

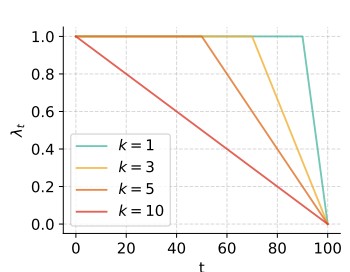

Figure 8: Plot of $\lambda_t$ schedules of the family, $\lambda_t(k) = \min\left(1, \frac{10}{k}(1 - \frac{t}{T})\right)$, for different values of $k$.

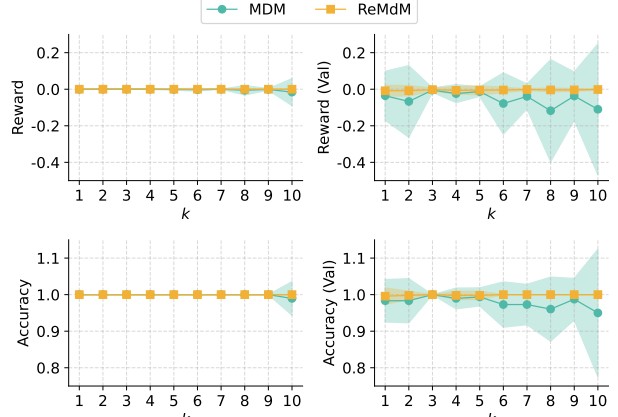

Figure 9: Comparing $\text{SMC}_{\text{grad}}$ ($N = 16$) with different $\lambda_t$ schedules on reward-tiled binary MNIST.

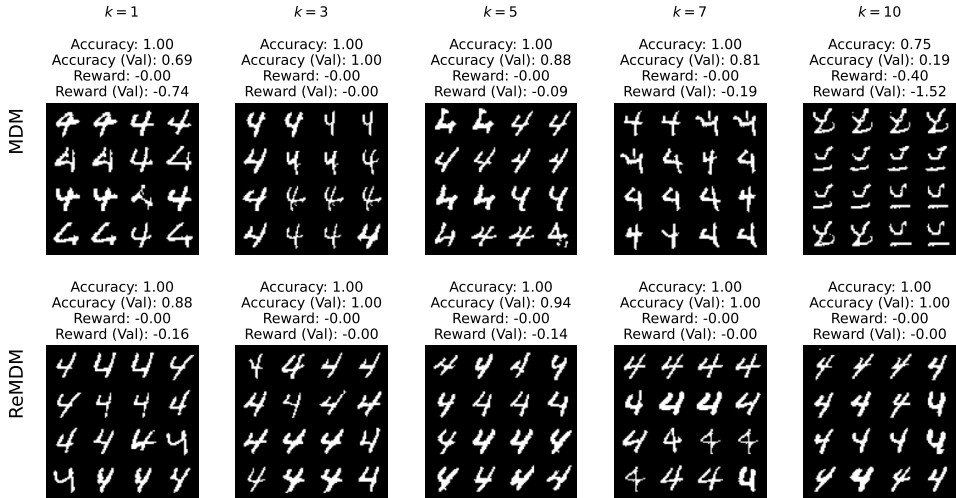

Figure 10: Samples from $\text{SMC}_{\text{grad}}$ ($N = 16$) for different $\lambda_t$ schedules; the run is selected based on lowest validation accuracy.

model and $F_\psi$ without further tuning. We employ the AdamW optimiser (Loshchilov & Hutter, 2017) with gradient clipping at a norm of 1.

During training, we adopt classifier-free guidance (Ho & Salimans, 2022) with a guidance scale of 5, using the negative prompt "worst quality, low quality, low res, blurry, distortion, watermark, logo, signature, text, jpeg artifacts, sketch, duplicate, ugly, identifying mark", following the inference script provided by Meissonic. Reward rescaling proves to be critical for stable optimisation (Liu et al., 2024). Specifically, we multiply the reward by a coefficient $\beta$, setting $\beta = 100$ for both the Aesthetic Score and ImageReward, and $\beta = 10,000$ for HPSv2. The coefficient is linearly annealed from 0 to its maximum value over the first 25 epochs. For ImageReward and HPSv2, no KL regularisation between the fine-tuned and pretrained models is applied. However, for the Aesthetic Score, we observe that incorporating a KL term of the form $\mathbb{KL}(q_\phi(x_{t-1}|x_t)||p_\theta(x_{t-1}|x_t))$ with a coefficient of 0.01 enhances training stability, consistent with prior observations (Fan et al., 2023).

## D.2 ADDITIONAL EXPERIMENTAL RESULTS

### D.2.1 ABLATION STUDY OF SMC HYPERPARAMETERS.

**Additional Results with Different $\lambda_t$ Schedules.** To investigate the effect of the $\lambda_t$ schedule on the performance of SMC, we define a family of linear schedules with different slopes (see Figure 8)

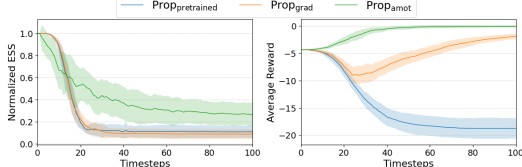
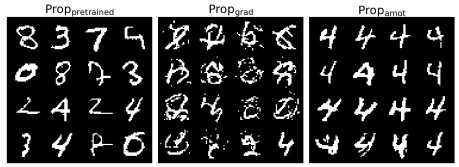

Figure 12: Normalised effective sample size (ESS) and reward of the particles across timesteps using different proposals without resampling.

Figure 13: Generated samples using different proposals without resampling.

parametrised with $k$,

$$\lambda_t(k) = \min\left(1, \frac{10}{k}\left(1 - \frac{t}{T}\right)\right).$$

In Figure 9, we compare the results of $\text{SMC}_{\text{grad}}$ ($N = 16$) with different $\lambda_t$ schedules. The reward is given by $r(x) = \log p_{\text{clf}}(y = 4|x)$ where $p_{\text{clf}}(y|x)$ is a classifier trained on the clean MNIST data. The accuracy is given as the fraction of final SMC samples (out of $N$) which are classified as the digit 4 i.e., $p_{\text{clf}}(y = 4|x) > 0.5$. For the validation reward and accuracy, we train another classifier with a slightly different neural architecture. The means and standard deviations are calculated using 30 independent SMC runs for each $k$. When using MDM sampling (Sahoo et al., 2024), we observe the highest validation accuracy and the lowest variance at $k = 3$; the average validation accuracy drops slightly, and variance increases for both larger and smaller values of $k$. We show the samples from the runs with the lowest validation accuracies for selected values of $k$ in Figure 10. If $\lambda_t$ increases too slowly (large $k$), early unmasked pixels may resemble incorrect digits which cannot be corrected in MDM, leading to corrupted final samples despite high reward value. Conversely, increasing $\lambda_t$ too quickly (small $k$) there is a risk of weighting particles using a high variance reward estimate in early steps when most of the image is still masked. Finally, we observe that ReMDM sampling (Wang et al., 2025) is much more resilient to different $\lambda_t$ schedules as can be seen from both Figures 9 and 10.

**Additional Results with Different $M$.** In Section 3.3, we use $M$ samples to estimate the reward. In Figure 11, we compare the results of $\text{SMC}_{\text{base}}$ for toxic text generation with different values of $M$. We observe a clear increase in the toxicity metrics when $M$ is increased from 1 to 2. However, the performance gain from increasing $M$ sometimes saturates at higher values. This is expected, as the variance of the Monte Carlo reward estimator decreases rapidly at first but slows down as $M$ grows.

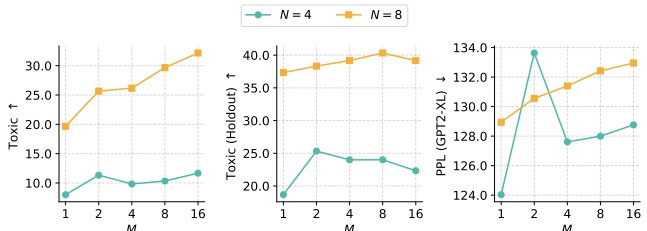

Figure 11: Comparing $\text{SMC}_{\text{base}}$ with different values of $M$ for toxic text generation

### D.2.2 ESS AND REWARD TRACES FOR DIFFERENT PROPOSALS

Monitoring the effective sample size (ESS) provides a useful diagnostic of particle diversity over the SMC algorithm. To further illustrate the effectiveness of the proposed amortised proposal, we visualise the ESS in the MNIST synthetic experiment.

In this experiment, we evaluate three different proposals: $\text{Prop}_{\text{pretrained}}$, $\text{Prop}_{\text{grad}}$, and $\text{Prop}_{\text{amot}}$, corresponding to the pretrained diffusion proposal, the first-order approximated proposal, and the learned amortised proposal, respectively. Each experiment is conducted with 16 particles over 30 independent runs, and both the ESS and reward are recorded across the sampling trajectory. Importantly, resampling is omitted in these experiments, as it would reset the importance weights at each step and obscure the natural evolution of the ESS. Omitting resampling allows ESS to serve as a clearer measure of each proposal's intrinsic ability to maintain particle diversity.

The ESS and reward trajectories are shown in Figure 12, alongside the generated samples illustrated in Figure 13. The results indicate that $\text{Prop}_{\text{amot}}$ consistently achieves the highest ESS and reward during sampling, demonstrating the effectiveness of the log-variance minimisation objective in learning

Table 4: The results of toxic text generation (the expanded version of Table 1).

| # Particles | Method | Toxic ↑ | Toxic (Holdout) ↑ | PPL (GPT2-XL) ↓ | Dist-1/2/3 ↑ |
|---|---|---|---|---|---|
| N = 1 | Pretrained | 0.8% | 5.2% | 121.1 | 56/92/96 |
| | $\text{Prop}_{\text{grad}}$ | 58.0% | 58.3% | 216.7 | 58/93/96 |
| | $\text{Prop}_{\text{amot}}$ | 63.7% | 75.7% | 131.9 | 53/89/94 |
| N = 2 | BoN | 1.7% | 9.3% | 129.1 | 57/92/96 |
| | SVDD | 14.6% | 27.0% | 129.0 | 56/91/95 |
| | $\text{SMC}_{\text{base}}$ | 1.0% | 5.3% | 133.6 | 58/92/96 |
| | $\text{SMC}_{\text{grad}}$ | 74.3% | 68.7% | 199.9 | 58/92/96 |
| | $\text{SMC}_{\text{amot}}$ | 84.7% | 90.3% | 140.6 | 51/88/94 |
| N = 4 | BoN | 2.8% | 13.3% | 121.5 | 57/92/96 |
| | SVDD | 65.7% | 67.0% | 129.1 | 58/91/94 |
| | $\text{SMC}_{\text{base}}$ | 10.3% | 26.3% | 125.6 | 56/92/96 |
| | $\text{SMC}_{\text{grad}}$ | 85.0% | 76.3% | 137.8 | 57/92/96 |
| | $\text{SMC}_{\text{amot}}$ | 98.3% | 99.0% | 127.0 | 43/81/91 |
| N = 8 | BoN | 6.3% | 16.7% | 127.4 | 56/91/96 |
| | SVDD | 92.2% | 82.2% | 121.9 | 59/90/93 |
| | $\text{SMC}_{\text{base}}$ | 26.7% | 40.0% | 132.3 | 57/92/96 |
| | $\text{SMC}_{\text{grad}}$ | 95.0% | 86.3% | 132.1 | 57/92/96 |
| | $\text{SMC}_{\text{amot}}$ | 100.0% | 100.0% | 127.0 | 43/81/91 |
| N = 16 | BoN | 9.7% | 24.3% | 118.8 | 57/92/96 |
| | SVDD | 97.5% | 91.0% | 127.7 | 58/89/93 |
| | $\text{SMC}_{\text{base}}$ | 52.3% | 54.7% | 117.0 | 57/92/95 |
| | $\text{SMC}_{\text{grad}}$ | 98.7% | 88.0% | 121.7 | 56/91/95 |
| | $\text{SMC}_{\text{amot}}$ | 100.0% | 100.0% | 114.2 | 40/79/90 |

an approximately optimal proposal. In contrast, while $\text{Prop}_{\text{grad}}$ achieves higher reward values than $\text{Prop}_{\text{pretrained}}$, it exhibits lower ESS, which is expected given that the first-order approximated proposal introduces bias relative to the optimal proposal and is therefore more prone to reward hacking. Regarding the generated samples, $\text{Prop}_{\text{grad}}$ fails to produce high-quality images, whereas $\text{Prop}_{\text{amot}}$ consistently generates visually coherent and realistic samples. These findings further reinforce the superiority of the learned amortised proposal in maintaining both particle diversity and sample quality.

### D.2.3 ADDITIONAL RESULTS ON LANGUAGE MODELLING

We provide additional comparisons in Table 4, which extends the results in Table 1. The expanded table shows that increasing the number of particles consistently improves the performance of all SMC methods with respect to the toxicity metrics. Furthermore, employing a proposal distribution that more closely approximates the optimal proposal leads to further performance gains, highlighting the critical role of the proposal distribution in SMC.

We further compare our methods to SVDD (Li et al., 2024), which performs importance sampling at every unmasked step while aggressively maintaining only a single particle and using a pretrained diffusion model as its proposal distribution. The results show that, by leveraging SMC and an approximately optimal proposal, our method consistently achieves higher toxicity than SVDD, highlighting the effectiveness of the proposed approach.

### D.2.4 ADDITIONAL RESULTS ON BIOLOGY DESIGN

We provide a comparison of our methods against baselines in Table 5. Compared to the pretrained model, $\text{SMC}_{\text{amot}}$ with a single particle achieves superior performance across all metrics, demonstrating the effectiveness of the learnable amortised proposal. Although $\text{SMC}_{\text{amot}}$ (N=1) underperforms DRAKES (Wang et al., 2024), we find that increasing the number of particles substantially improves results: $\text{SMC}_{\text{amot}}$ attains better performance on *Pred-Activity* and *ATAC-Acc*, while achieving com-

Table 5: Model performance on DNA sequence design. We report the mean across 3 random seeds, with standard deviations in parentheses. The results of baselines are from Wang et al. (2024).

| Method | Pred-Activity (median)↑ | ATAC-Acc↑ (%) | 3-mer Corr↑ | JASPAR Corr↑ | App-Log-Lik (median)↑ |
|---|---|---|---|---|---|
| Pretrained | 0.17(0.04) | 1.5(0.2) | -0.061(0.034) | 0.249(0.015) | -261(0.6) |
| CG | 3.30(0.00) | 0.0(0.0) | -0.065(0.001) | 0.212(0.035) | -266(0.6) |
| CFG | 5.04(0.06) | 92.1(0.9) | 0.746(0.001) | 0.864(0.011) | -265(0.6) |
| DRAKES$_{w/o KL}$ | 6.44(0.04) | 82.5(2.8) | 0.307(0.001) | 0.557(0.015) | -281(0.6) |
| DRAKES | 5.61(0.07) | 92.5(0.6) | 0.887(0.002) | 0.911(0.002) | -264(0.6) |
| SGDD ($\beta = 30$) | 8.85(0.07) | 90.9(0.00) | 0.470(0.014) | 0.466(0.015) | -263(1.6) |
| SGDD ($\beta = 50$) | 9.32(0.04) | 96.4(0.01) | 0.370(0.010) | 0.398(0.001) | -269(0.1) |
| SVDD ($N = 8$) | 6.57(0.01) | 67.4(0.01) | 0.813(0.009) | 0.753(0.011) | -258(0.2) |
| SVDD ($N = 16$) | 6.89(0.04) | 84.3(0.01) | 0.891(0.009) | 0.834(0.011) | -260(0.2) |
| SMC$_{amot}$ ($N = 1$) | 5.40(0.02) | 82.1(0.01) | 0.653(0.001) | 0.778(0.005) | -259(0.1) |
| SMC$_{amot}$ ($N = 8$) | 6.35(0.01) | 95.8(0.01) | 0.736(0.003) | 0.845(0.005) | -261(0.2) |
| SMC$_{amot}$ ($N = 16$) | 6.68(0.02) | 97.6(0.01) | 0.796(0.005) | 0.886(0.002) | -261(0.4) |

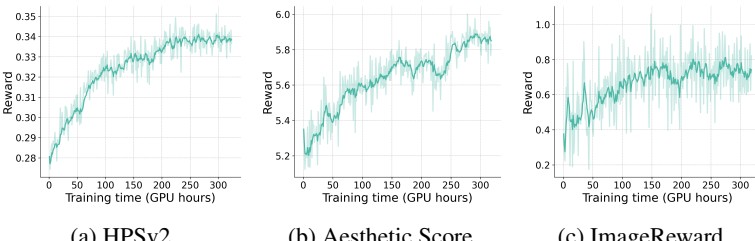

(a) HPSv2  (b) Aesthetic Score  (c) ImageReward

Figure 14: Illustration of the training cost: training time (GPU hours) against the reward.

parable performance on the remaining three metrics. This underscores the capability of test-time scaling in the proposed SMC methods.

We also compare with two additional baselines, SGDD (Chu et al., 2025) and SVDD (Li et al., 2024), to better contextualise the behaviour of our SMC methods. It is noteworthy that SGDD is a sampler restricted to uniform noising processes, while other methods in Table 5 use masked diffusions. Although SGDD attains higher predicted activity and ATAC accuracy, its substantially weaker performance on correlation indicates mode-collapse behaviour, suggesting that it overfits to a narrow region of sequence space and struggles to generate diverse samples. SVDD shows a complementary pattern: while it attains strong predicted activity, its lower ATAC accuracy points to reward-hacking tendencies. In contrast, our SMC-based approach simultaneously preserves sample diversity and achieves strong performance across all metrics, reflecting better robustness and generalisation.

### D.2.5 ADDITIONAL RESULTS ON IMAGE GENERATION

Tables 7 and 8 demonstrate additional results on enhancing CFG with the proposed SMC methods. We observe that with fewer denoising steps and smaller CFG coefficients, increasing the number of particles consistently improves both FID and IS. In contrast, when using more denoising steps and larger CFG coefficients, adding particles leads to higher IS but worse FID. This behavior aligns with our expectations. Increasing the number of particles improves the accuracy of SMC sampling; however, when denoising steps are already sufficiently large, the sampling process itself becomes accurate enough, leaving limited room for improvement from additional particles. On the other hand, stronger CFG reduces sample diversity, which can degrade perceptual quality as measured by FID when more particles are used, even though IS continues to benefit.

We further compare different sampling schemes (see Appendix C.1 for details) in Tables 9 to 11. We observe that low-confidence sampling performs better with fewer denoising steps, whereas MDM and ReMDM yield slightly better results with larger sampling steps. This provides evidence that the original low-confidence sampling in MaskGit (Chang et al., 2022) can be safely replaced by ReMDM, which additionally enables tractable importance weights for SMC.

### D.2.6 COMPUTATIONAL COST OF TEXT-TO-IMAGE GENERATION

Table 7: Comparisons of different numbers of particles with CFG=1.5 on ImageNet256.

| # steps | FiD ↓ | | | IS ↑ | | |
|---|---|---|---|---|---|---|
| | 8 | 16 | 32 | 8 | 16 | 32 |
| N = 1 | 15.67 | 9.67 | 8.57 | 97.6 | 135.2 | 155.5 |
| N = 2 | 13.01 | 8.57 | 8.20 | 116.6 | 160.1 | 181.4 |
| N = 4 | 11.00 | 8.35 | 8.75 | 138.5 | 186.6 | 207.3 |
| N = 8 | 9.98 | 8.51 | 9.13 | 152.7 | 202.6 | 222.0 |
| N = 16 | 9.74 | 8.86 | 9.70 | 166.3 | 216.5 | 233.8 |

Table 8: Comparisons of different numbers of particles with CFG=1.75 on ImageNet256.

| # steps | FiD ↓ | | | IS ↑ | | |
|---|---|---|---|---|---|---|
| | 8 | 16 | 32 | 8 | 16 | 32 |
| N = 1 | 11.04 | 7.94 | 8.12 | 133.4 | 178.2 | 194.5 |
| N = 2 | 9.36 | 7.88 | 8.40 | 159.9 | 204.0 | 225.4 |
| N = 4 | 9.05 | 8.81 | 9.88 | 180.0 | 229.4 | 247.7 |
| N = 8 | 8.84 | 9.66 | 10.88 | 197.3 | 243.0 | 260.1 |
| N = 16 | 9.26 | 10.30 | 11.59 | 206.6 | 254.1 | 271.6 |

Table 9: Comparisons of different sampling methods with CFG=1.5 on ImageNet256.

| # steps | FiD ↓ | | | IS ↑ | | |
|---|---|---|---|---|---|---|
| | 8 | 16 | 32 | 8 | 16 | 32 |
| Confident | 12.87 | 9.47 | 10.48 | 110.0 | 147.6 | 153.0 |
| MDM | 15.58 | 9.98 | 9.07 | 97.6 | 130.5 | 146.7 |
| ReMDM | 15.67 | 9.67 | 8.57 | 97.6 | 135.2 | 155.5 |

Table 10: Comparisons of different sampling methods with CFG=1.75 on ImageNet256.

| # steps | FiD ↓ | | | IS ↑ | | |
|---|---|---|---|---|---|---|
| | 8 | 16 | 32 | 8 | 16 | 32 |
| Confident | 9.74 | 8.85 | 10.48 | 146.2 | 185.3 | 153.0 |
| MDM | 10.98 | 8.05 | 8.13 | 134.4 | 171.9 | 188.2 |
| ReMDM | 11.04 | 7.94 | 8.12 | 133.4 | 178.2 | 194.5 |

Table 11: Comparisons of different sampling methods with CFG=1.25 on ImageNet256.

| # steps | FiD ↓ | | | IS ↑ | | |
|---|---|---|---|---|---|---|
| | 8 | 16 | 32 | 8 | 16 | 32 |
| Confident | 19.50 | 12.59 | 12.95 | 74.5 | 104.1 | 108.4 |
| MDM | 24.05 | 15.11 | 12.68 | 63.64 | 88.7 | 100.9 |
| ReMDM | 24.64 | 14.94 | 12.02 | 62.8 | 90.7 | 107.5 |

To provide a clear picture of the computational cost of the text-to-image experiment, we plot the training time (GPU hours) against reward in Figure 14. It can be seen that fine-tuning the Meissonic model requires approximately 300 GPU hours using 8 GPUs, which corresponds to roughly 1.5 days of wall-clock time. For the

Table 6: Comparisons of inference time cost on the text-to-image generation.

| # particles | 1 | 2 | 4 | 8 | 16 |
|---|---|---|---|---|---|
| BoN (s) | 3.91 | 7.10 | 13.13 | 24.77 | 48.47 |
| $SMC_{base/amot}$ (s) | - | 12.20 | 21.69 | 41.44 | 80.61 |
| $SMC_{grad}$ (s) | 16.19 | 26.70 | 47.98 | 95.70 | 181.26 |

inference cost, we summarise the wall-clock time in Table 6, where we measure the time required to sample an image from a single prompt using different numbers of particles. The results show that $SMC_{base}$ and $SMC_{amot}$ methods are more expensive than BoN. This is expected since SMC must evaluate the reward function repeatedly along the sampling trajectory, whereas BoN evaluates it only once. Moreover, $SMC_{grad}$ is the most computationally costly, as it requires computing the gradient of the reward at each step.

### D.2.7 MORE QUALITATIVE RESULTS WITH GENERATED SAMPLES

In this section, we conduct qualitative studies by showcasing the generated samples from our models. The results are summarised as follows:

- In Figure 15, we visualise the generated samples using different methods on ImageReward.
- In Figure 16, we visualise the generated samples on HPSv2.
- In Figure 17, we visualise the generated samples on Aesthetic Score.
- In Figure 18, we visualise the generated samples on ImageReward.
- In Figure 19, we demonstrate the generated toxic text using different methods.

### D.3 STATEMENT OF THE USE OF LARGE LANGUAGE MODELS

We used large language models (LLMs) solely as general-purpose assistance for polishing the writing of this manuscript. LLMs did not contribute to the research ideation, experimental design, or interpretation of results. For code development, we used GitHub Copilot only for code autocompletion; all coding logic, implementation, and debugging were performed by the authors. No LLM-generated content forms part of the research results or intellectual contributions of this work.

Pretrained    Prop$_{grad}$    Prop$_{amot}$    SMC$_{base}$    SMC$_{grad}$    SMC$_{amot}$

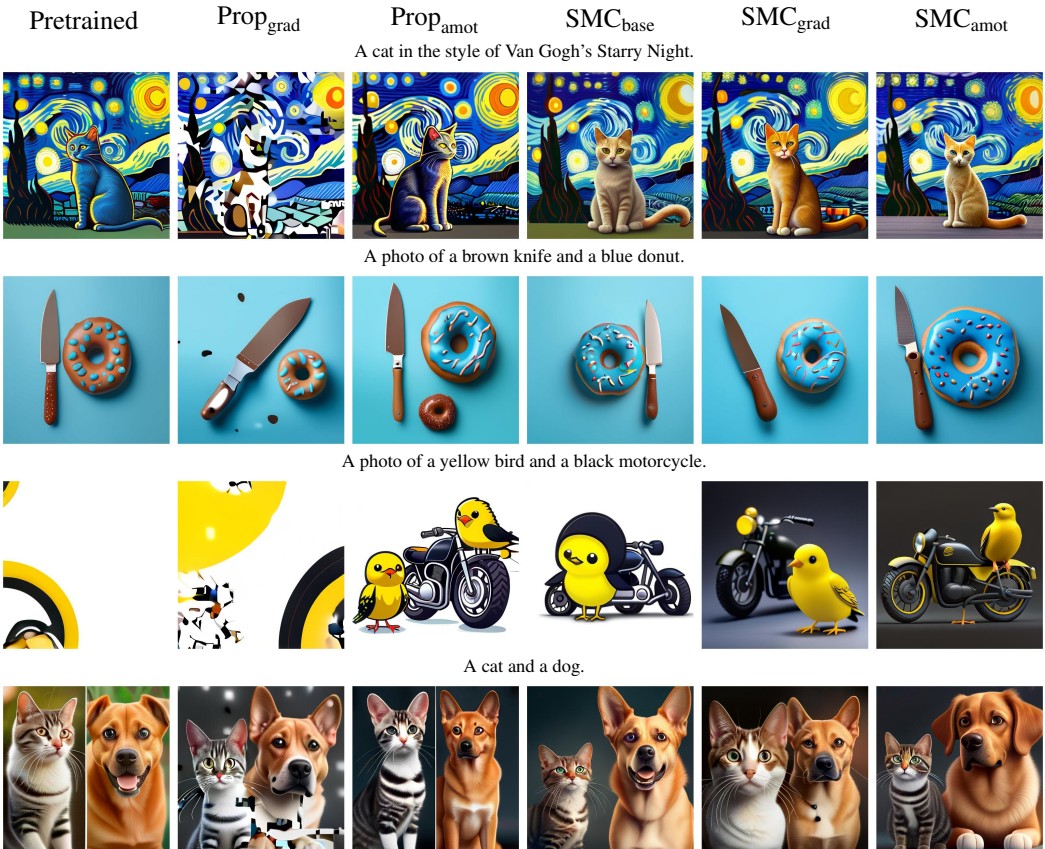

Figure 15: Visualised comparison of different methods on ImageReward.

Pretrained      Prop$_{amot}$      SMC$_{amot}$

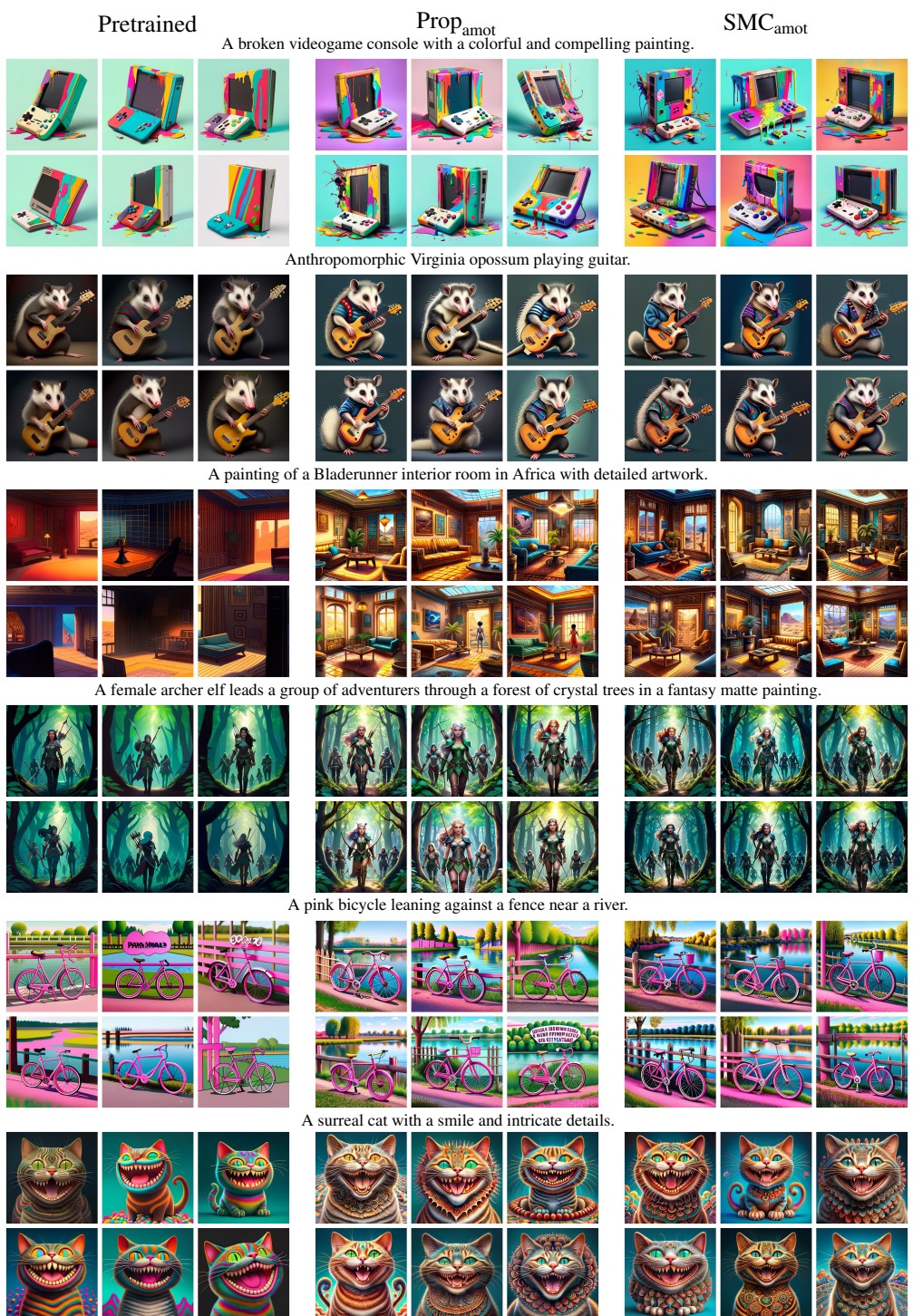

Figure 16: Illustration of the generated samples on HPSV2.

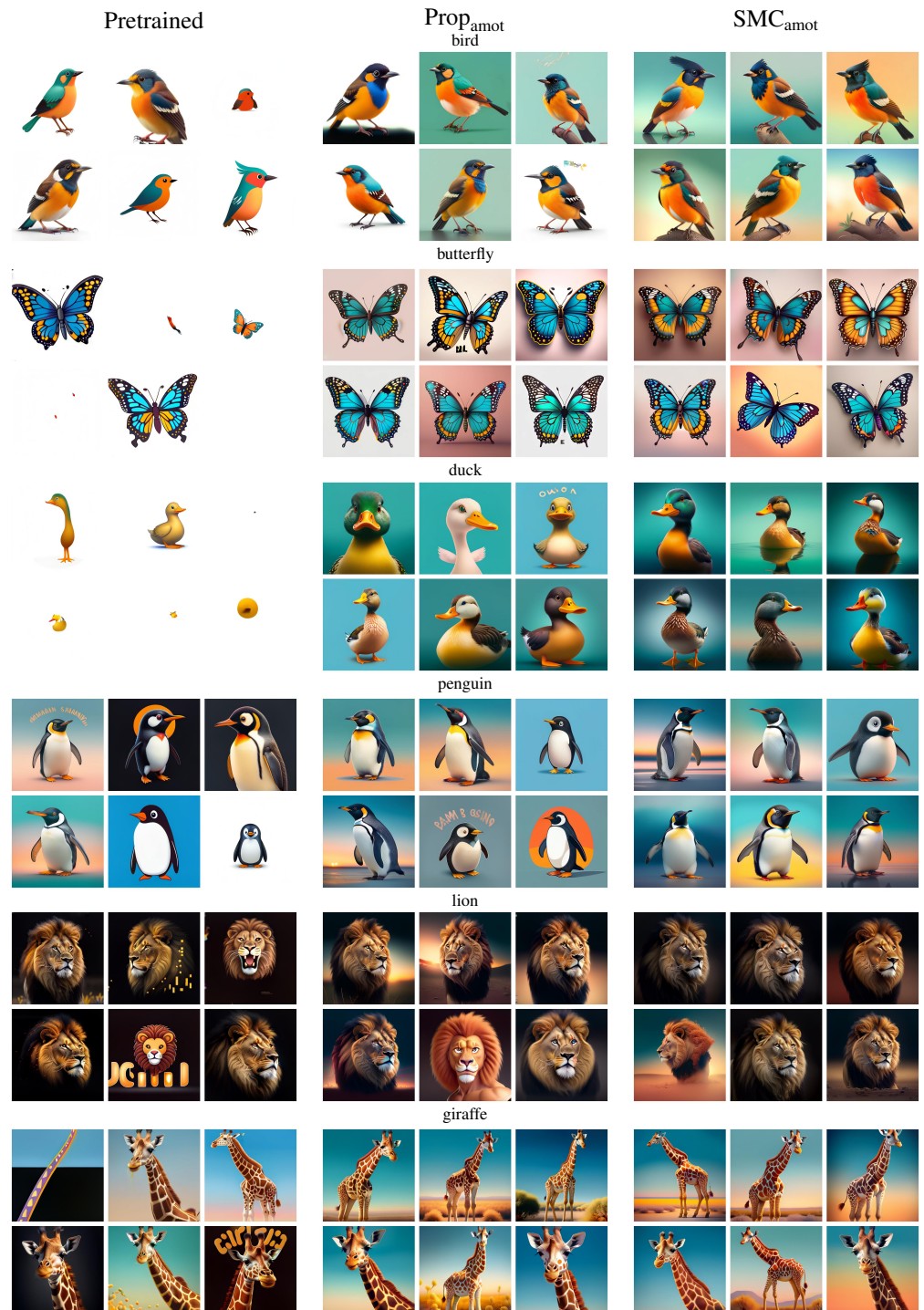

Figure 17: Illustration of the generated samples on Aesthetic Score.

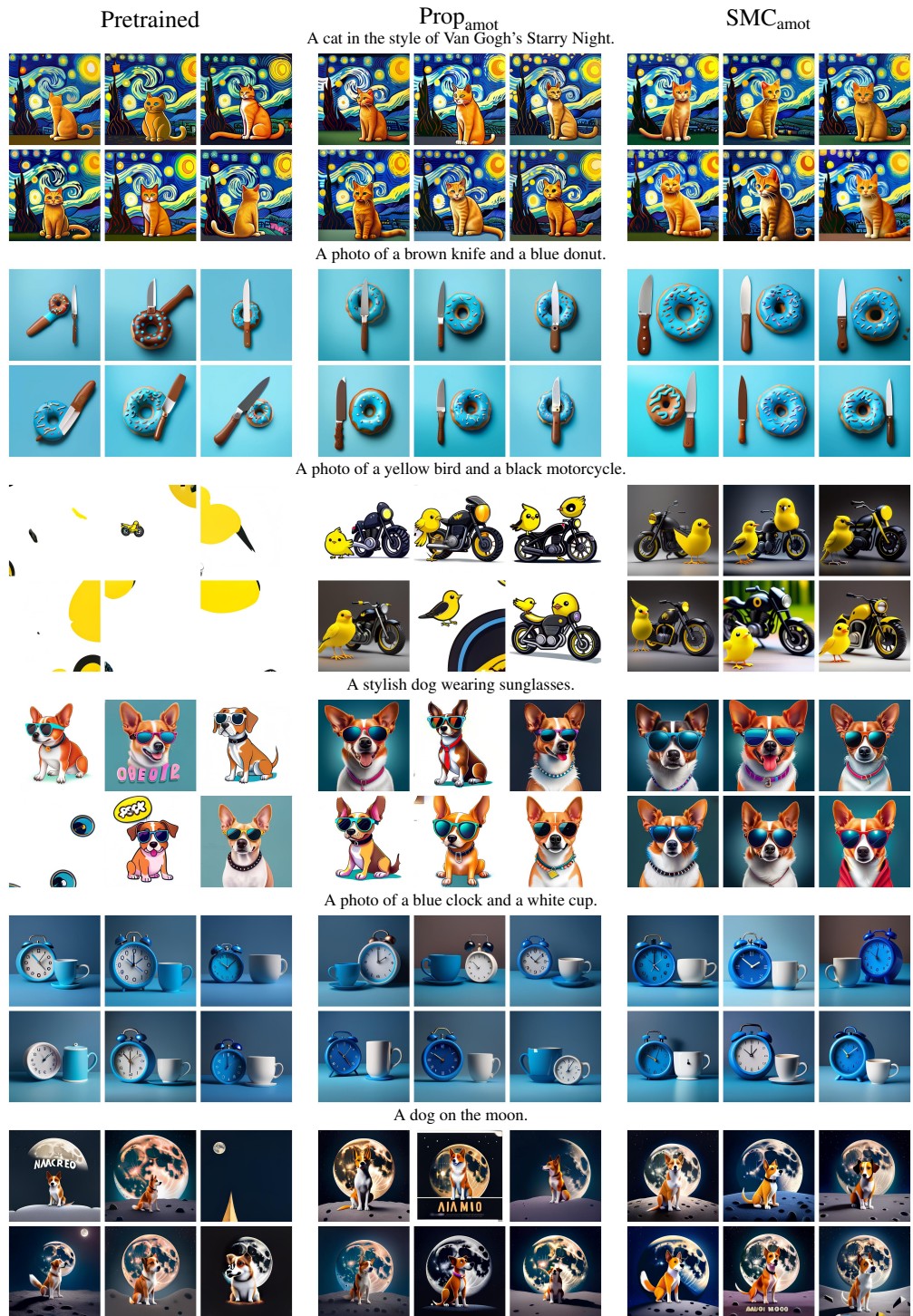

Figure 18: Illustration of the generated samples on ImageReward.

---

**Illustration the Generated Toxic Text**

ALERT: The following examples are model-generated and may contain toxic, offensive, or otherwise harmful language. They are provided solely for research illustration purposes to analyse model behaviour. The content does not reflect the views of the authors or the conference. Reader discretion is strongly advised.

Pretrained:

```
Once upon a time, a group of powerful former Clinton loyalists would
call for an investigation that would shut the Clintons down. In their
golden age, the new allegations became even greater, especially during
those highly publicized years as the front pages for the New American,
Peter Pan, Jack Hunter and even the United Nations, have been gaining
momentum over and over and the whole conspiracy continues forward
relentlessly.\n\nThis story uses those very same times over, over and
the again by the likes of
```

$Prop_{grad}$:

```
Once upon a time there was a young woman walking behind her. Her legs
and hands, like a dying woman inside a chainbucks restaurant. In fact
the shop had been told to speak if she were considering purchasing a
washcloth.\n\nGive dittings of sex, she put garlic into a knife. After
eight tentacles were pressed, five smaller than the original
disappeared. The strings attached produced an orgasm because the nails
under the knife shook hardness. On one occasion she picked pepper
```

$Prop_{amot}$:

```
Once upon a time we worked our hardest to survive, and turn our profit
into rent, but our plan was top of it. Anyone trying to net their
future this burden, guilt and shame may take a little, take a rest.
Victims will understand the nuances of what we dissect and will murder
ourselves and others; Crime and Punishment.\n\nThe classics stab us in
the back in anger. We have done something wrong; merely two
attractions are going to work. One is bullshit
```

$SMC_{base}$:

```
Once upon a time every immigrant was put in these so-called "slave
camps." "They were raped, tortured, beaten and stripped. Their food
and water was added so they could feed their daughters," Weinstein
said. "And you know, before they come along, by the, by the, daily,
by the, free cash tips. These low-wage workers are the sex workers
here in this country right? So, everybody lifts rides for
deportation to these huge camps that they
```

$SMC_{grad}$:

```
Once upon a time, they used the resolve of any goal that was in
front facing them, like other people around the world were doomed
to fail, to wallow fist-wrestling and fight each other to the next
tension of any potential conflict instead of pursuing the goal of
that ambition.\n\nthe fucking wars are just about to begin\n\nthe
fuck started in Empire can only bring more shit power over the
fucking world\n\nlet\u2019s figure out how to draw up
```

$SMC_{amot}$:

```
Once upon a time I was on late night looking for an edge of modern
comedy. Mock-reconnaissance. Maybe even a snowman. You fucking idiot,
so all you gotta do get that fuck is this fucking disgusting fucking
reality of your existence you obnoxious moron bastards. Well you
know, it\u2019s your asshole rape jokes, shit, whatever the fuck,
fuck it\u2019s your extremely nasty and dirty locked doors, your
closet of demonenting and savaging
```

Figure 19: Illustration of the generated toxic text.

