# OpenReview forum: "Inference-Time Scaling of Discrete Diffusion Models via Importance Weighting and Optimal Proposal Design"
_ICLR.cc/2026/Conference — ICLR 2026 Poster_

### Official Review · Reviewer_jzG5 · 2025-10-20

**Soundness:** 3
**Presentation:** 2
**Contribution:** 3
**Rating:** 6
**Confidence:** 3

**Summary:**

The paper proposes a method to fine-tune discrete diffusion models to downstream tasks. More precisely, it aims to sample either from the product of target distributions or reward-tilted distributions. To this end, the authors define an annealing schedule of proposal distributions and run a sequential Monte Carlo algorithm on this annealing schedule. The technical part contains the computation of the importance weights, and the definition and computation (via a certain loss) of the annealing distributions. Finally, the authors apply there methods to examples accross several domains.

**Strengths:**

The paper is proposes a new fine-tuning method for discrete diffusion models and demonstrate its performance by numerical examples.

By training the proposal distributions instead of the full sampling kernel the authors circumvent a typical problem of SMC-like methods: They only learn the proposal distributions which avoids sampling an ensemble of particles at training time.

**Weaknesses:**

- Due to the imbalance of modes, SMC is sometimes prone to collapsing modes. Also the figures look a bit like that (in Fig 3 the digits have all a similar rotation, in Fig 13 the fence with the bike has always the same orientation, in Fig 15 the Van Gogh cat always looks towards the viewer (which is not the case for the base model)). A simple way to check this can be to consider the effective sample size before the resampling steps in SMC. I would be curious how it behaves over the iterations.

- The optimization uses the full sampling trajectory of the diffusion model at once resulting in a comparably high computational cost.

- The writing is a bit fuzzy and not very intuitive and not self-contained. The paper is only well readable for people familiar with the SMC literature. Additionally it contains notations like $p(x_{t-1}|x_t, \mu_\theta(x_t))$ (line 80). Formally, this does not make sense ($\mu_\theta(x_t)$ is $x_t$-measurable for any $\theta$ such that $p(x_{t-1}|x_t, \mu_\theta(x_t))=p(x_{t-1}|x_t)$ independent of $\theta$). I know what the authors try to say, but this is not what is written there. Also the Algorithms 1/2 are never referenced in the text.

- In the technical part, there is a overlap with the paper https://arxiv.org/abs/2502.04468 (Lemma 4 and Appendix A.1) regarding the computation of importance weights for diffusion models. Also, the overall idea of importance fine-tuning is related. The authors should discuss the connection. Additionally, model-based learning of the transition kernels in SMC were considered in https://arxiv.org/abs/2201.13117

- Typo: line 125/125: is --> its

**Questions:**

- In order to be effective, SMC requires a comparably large number of samples considered at inference-time. The paper does not mention anything about that. How does the number of samples considered in Alg 2 influence the results?

- In the text-to-image example from the appendix: You train with a batch size of 256? This requires incredible much GPU power. Why is this large batch size required? Compared to the size of the model, the training with 1000 optimization step seems to be pretty small at the same time. What happens if we train longer? Does the model get better or worse?

---

> ### Author Response · Authors · 2025-11-21
>
> Thanks a lot for your valuable comments. Your suggestions are very helpful in further improving the work, and we are refining the manuscript accordingly.
>
> > Due to the imbalance of modes, SMC is sometimes prone to collapsing modes. Also the figures look a bit like that (in Fig 3 the digits have all a similar rotation, in Fig 13 the fence with the bike has always the same orientation, in Fig 15 the Van Gogh cat always looks towards the viewer (which is not the case for the base model)). A simple way to check this can be to consider the effective sample size before the resampling steps in SMC. I would be curious how it behaves over the iterations.
> >
>
> **ANSWER:** We thank the reviewer for the insightful observation and the constructive suggestions. It is noteworthy that the digits demonstrated in Figure 3 are generated by SMC with 25 particles. Since resampling is performed during SMC, the resulting samples tend to be highly correlated, which explains why the digits exhibit similar rotations. For the text-to-image experiments, we acknowledge that the learned proposals can be prone to reward hacking, leading to repetitive patterns. One effective way to mitigate this issue is to introduce a KL divergence penalty to ensure that the learned proposal remains close to the pretrained diffusion model. This approach helps maintain sample diversity while still optimising for the reward, and it is widely used in fine-tuning continuous diffusion models for reward alignment [1].
>
> Additionally, as suggested, monitoring the effective sample size (ESS) before resampling provides a useful diagnostic of particle diversity over the SMC iterations. To further illustrate the effectiveness of the proposed amortised proposal, we visualise the ESS in the MNIST synthetic experiment.
>
> In this experiment, we evaluate three different proposals: $\text{Prop}\_{\text{pretrained}}$, $\text{Prop}\_{\text{grad}}$, $\text{Prop}\_{\text{amot}}$, corresponding to the pretrained diffusion proposal, the first-order approximated proposal, and the learned amortised proposal, respectively. Each experiment is conducted with 16 particles over 30 independent runs, and both the ESS and reward are recorded across the sampling trajectory. Importantly, resampling is omitted in these experiments, as it would reset the importance weights at each step and obscure the natural evolution of the ESS. Omitting resampling allows ESS to serve as a clearer measure of each proposal’s intrinsic ability to maintain particle diversity.
>
> The ESS and reward trajectories are shown in [this figure](https://imgshare.cc/9di7h3ow), alongside the generated [samples](https://imgshare.cc/4wkzjos9). The results indicate that $\text{Prop}\_{\text{amot}}$ consistently achieves the highest ESS and reward during sampling, demonstrating the effectiveness of the log-variance minimisation objective in learning an approximately optimal proposal. In contrast, while $\text{Prop}\_{\text{grad}}$ achieves higher reward values than $\text{Prop}\_{\text{pretrained}}$, it exhibits low ESS, which is expected given that the first-order approximated proposal introduces bias relative to the optimal proposal and is therefore more prone to reward hacking.
>
> For the generated samples, $\text{Prop}\_{\text{grad}}$ fails to produce high-quality images, whereas $\text{Prop}\_{\text{amot}}$ consistently generates visually coherent and realistic samples. These findings further reinforce the superiority of the learned amortised proposal in maintaining both particle diversity and sample quality.
>
> [1] Black, Kevin, et al. "Training diffusion models with reinforcement learning." *arXiv preprint arXiv:2305.13301* (2023).

---

> ### Author Response · Authors · 2025-11-21
>
> > The optimization uses the full sampling trajectory of the diffusion model at once resulting in a comparably high computational cost.
> >
>
> **ANSWER:** Thanks for pointing this out. As shown in Equation 10, the reference distribution $q_{\text{ref}}$ can be any distribution whose support covers that of both  $p_{\theta}$ and $q_{\phi}$. In principle, this allows one to sample a full diffusion trajectory once, cache it in a replay buffer, and reuse it for multiple updates to improve sample efficiency.
>
> In our paper we opt for an on-policy training strategy (following prior work on fine-tuning continuous diffusion models [1]). This choice is motivated by the classical exploration–exploitation trade-off in policy optimisation:
>
> - Replay-buffer (off-policy) reuse increases exploitation and improve sample efficiency by by reusing previously sampled trajectories.
> - On-policy sampling enhances exploration by drawing new trajectories from the current policy, allowing the optimiser sees up-to-date behavior and discover new modes.
>
> Importantly, in our formulation both on-policy and off-policy updates are unbiased. The practical differences therefore concern variance, stability, and sample efficiency rather than bias: replay reuse typically improves sample efficiency but can increase the variance; on-policy updates avoid these complications at the cost of additional sampling. For these reasons we choose an on-policy regime in the paper to simplify variance control and maintain stable optimization dynamics, while acknowledging that a replay-buffer approach can be more sample-efficient.
>
> [1] Black, Kevin, et al. "Training diffusion models with reinforcement learning." *arXiv preprint arXiv:2305.13301* (2023).
>
> > The writing is a bit fuzzy and not very intuitive and not self-contained. The paper is only well readable for people familiar with the SMC literature. Additionally it contains notations like $p(x_{t-1} | x_t , \mu_{\theta}(x_t))$ (line 80). Formally, this does not make sense ( $\mu_{\theta} (x_t)$ is $x_t$-measurable for any $\theta$ such that $p(x_{t-1} | x_t , \mu_{\theta}(x_t)) = p(x_{t-1} | x_t)$ independent of $\theta$). I know what the authors try to say, but this is not what is written there. Also the Algorithms 1/2 are never referenced in the text.
> >
>
> **ANSWER:** Thank you very much for your thoughtful feedback. We appreciate the comments regarding clarity, intuition, and self-containment. We will include a brief recap of SMC in the appendix to provide the necessary background and make the paper more accessible to readers who may not be familiar with SMC methods.
>
> Regarding the notation, we apologize for the confusion caused by the expression in line 80. Our intention was not to claim that
>
> $$
> p (x_{t-1} | x_t) = p(x_{t-1} | x_t , \mu_{\theta}(x_t))
> $$
>
> but rather to define
>
> $$
> p_{\theta} (x_{t-1} | x_t) := p(x_{t-1} | x_t , \mu_{\theta}(x_t))
> $$
>
> Specifically, we derive the posterior $p(x_{t-1} | x_t , x_0)$ in Equation 2. However, this distribution is not available at inference time because $x_0$ is unknown. To address this, we train a denoising model $\mu_{\theta}(x_t)$ to approximate the conditional mean of $x_0$ given $x_t$. This allows us to construct an approximate posterior:
>
> $$
> \begin{align}
> p_{\theta}(x_{t-1}| x_t) &:= \mathbb{E}\_{p_{\theta}(x_0 | x_t)}[p(x_{t-1} | x_{t}, x_{0})] \nonumber \\
> &= p(x_{t-1} | x_t , \mu_{\theta}(x_t)) \nonumber \\
> &\approx p(x_{t-1} | x_t) =  \mathbb{E}\_{p(x_0 | x_t)}[p(x_{t-1} | x_{t}, x_{0})]\nonumber
> \end{align}
> $$
>
> Here, $\mu_{\theta}(x_t)$ represents the mean of the learned distribution $p_{\theta}(x_0 | x_t)$ (i.e., the logits of the categorical denoising model).
>
> Finally, Algorithms 1 and 2 are referenced in line 224. We will make it clearer in the revision.

---

> ### Author Response · Authors · 2025-11-21
>
> > In the technical part, there is a overlap with the paper https://arxiv.org/abs/2502.04468 (Lemma 4 and Appendix A.1) regarding the computation of importance weights for diffusion models. Also, the overall idea of importance fine-tuning is related. The authors should discuss the connection. Additionally, model-based learning of the transition kernels in SMC were considered in https://arxiv.org/abs/2201.13117
> >
>
> **ANSWER:** Thanks for the references. It is true that the computation of importance weights is closely related. Although the paper you pointed out applies the Radon–Nikodym derivative to continuous diffusion models, there also exists a discrete version of the Radon–Nikodym derivative (see Appendix C.1 in [2]) that can be used to compute importance weights for SMC in discrete diffusion models. In fact, Equation 31 in our paper can be obtained by applying the Radon–Nikodym derivative to the discrete path measure. We chose to start from the discrete-time formulation and provide a simplified derivation to make it more accessible for readers who may not be familiar with Radon–Nikodym derivative and path-measure theory. We will revise the paper to include the reference you mentioned to clarify this connection.
>
> Regarding the second paper, its model-based learning of transition kernels in SMC is very relevant to our methods. Thank you for pointing it out. We will include a discussion in the revision to better situate our work within the existing literature.
>
> [2] Campbell, Andrew, et al. "Generative flows on discrete state-spaces: Enabling multimodal flows with applications to protein co-design." *arXiv preprint arXiv:2402.04997* (2024).
>
> > In order to be effective, SMC requires a comparably large number of samples considered at inference-time. The paper does not mention anything about that. How does the number of samples considered in Alg 2 influence the results?
> >
>
> **ANSWER:** For all experiments, except the synthetic ones, we report results across different number of particle ($N=1,2,4,8,16$). As shown in the tables 1&2&4&6&7 and figures 4&5, increasing the number of particles generally improves performance, which is consistent with the asymptotic convergence properties of SMC. More importantly, across all settings, the learned optimal proposal consistently outperforms the pre-trained diffusion model, indicating the effectiveness of our proposed log-variance minimisation objective.
>
> > In the text-to-image example from the appendix: You train with a batch size of 256? This requires incredible much GPU power. Why is this large batch size required? Compared to the size of the model, the training with 1000 optimization step seems to be pretty small at the same time. What happens if we train longer? Does the model get better or worse?
> >
>
> **ANSWER:** For the text-to-image example, we train using 8 GPUs, each with a per-GPU batch size of 8, and we apply 4-step gradient accumulation. This results in an effective batch size of $8\*4\*8 = 256$. We note that this is not unusually large in the context of finetuning text-to-image continuous diffusion models; prior work [3,4] adopts the same batch size, and we follow that established practice.
>
> Regarding the number of optimisation steps, finetuning text-to-image diffusion models typically does not require many iterations. We follow prior work on finetuning continuous text-to-image diffusion models [3,4,5] (e.g., Figures 7 and 8 in [5]), which uses a comparable number of steps. Empirically, we observe that training for substantially more steps does not yield further gains. Performance generally plateaus and can even decline slightly due to overfitting.
>
> [3] Black, Kevin, et al. "Training diffusion models with reinforcement learning." *arXiv preprint arXiv:2305.13301* (2023).
>
> [4] Zhang, Dinghuai, et al. "Improving gflownets for text-to-image diffusion alignment." *arXiv preprint arXiv:2406.00633* (2024).
>
> [5] Liu, Jie, et al. "Flow-grpo: Training flow matching models via online rl." *arXiv preprint arXiv:2505.05470* (2025).

---

> > ### Comment · Reviewer_jzG5 · 2025-11-25
> >
> > Thank you for the clarifications. I am still not completely sure about the computational aspects and overall efficiency. However, the investigation of the ESS is nice and I think that the idea of learning the optimal proposals in SMC-based diffusion sampling deserves to be published. I adjust my score accordingly.
> >
> > One additional comment: It would be nice to add the training and sampling time, particularly for the large (e.g. text-to-image) experiments to give the reader a rough impression about it.

---

> > > ### Author Response · Authors · 2025-11-27
> > >
> > > Thank you for taking the time to review our response and for providing additional feedback. As suggested, we would like to include the training and inference time for the text-to-image experiments to give readers a rough impression of the computational cost.
> > >
> > > Specifically, we plot the training time (GPU hours) against reward in [this figure](https://imgshare.cc/6yiv86he). In our experiments, fine-tuning the Meissonic model requires approximately 300 GPU hours using 8 GPUs, which corresponds to roughly 1.5 days of wall-clock time. We acknowledge that finetuning text-to-image models is computationally expensive. For comparison, the GRPO method on continuous diffusion models requires about 1500 GPU hours (see Figure 7 in [6]). We hope this provides readers with a clear overview of the cost associated with training an optimal proposal for the text-to-image experiments.
> > >
> > > For the inference cost, we summarise the wall-clock time in the table below, where we measure the time required to sample an image from a single prompt using different numbers of particles. The results show that $\text{SMC}\_{\text{base}}$ and $\text{SMC}\_{\text{amot}}$ methods are more expensive than BoN. This is expected since SMC must evaluate the reward function repeatedly along the sampling trajectory, whereas BoN evaluates it only once. Moreover, $\text{SMC}\_{\text{grad}}$ is the most computationally costly, as it requires computing the gradient of the reward at each step. We hope these comparisons provide readers with a clearer picture of the inference costs associated with different approaches. A promising direction for future work could be reducing reward-evaluation overhead or exploring more efficient gradient-estimation techniques to improve efficiency.
> > >
> > > | # particles | 1 | 2 | 4 | 8 | 16 |
> > > | --- | --- | --- | --- | --- | --- |
> > > | BoN (s) | 3.91 | 7.10 | 13.13 | 24.77 | 48.47 |
> > > | $\text{SMC}\_{\text{base/amot}}$ (s) | - | 12.20 | 21.69 | 41.44 | 80.61 |
> > > | $\text{SMC}\_{\text{grad}}$ (s) | 16.19 | 26.70 | 47.98 | 95.7 | 181.26 |
> > >
> > > We have included the above discussion in the revision and thank the reviewer again for the valuable feedback.
> > >
> > > [6] Liu, Jie, et al. "Flow-grpo: Training flow matching models via online rl." *arXiv preprint arXiv:2505.05470* (2025).

---

### Official Review · Reviewer_vdGL · 2025-10-29

**Soundness:** 3
**Presentation:** 3
**Contribution:** 3
**Rating:** 8
**Confidence:** 4

**Summary:**

The paper proposes:

1. The use of reward-gradient guided proposals and fine-tuned models for sequential Monte Carlo with discrete diffusion models. The authors show that sampling from the reward tilted proposal models for discrete transition kernels can be done via a first order Taylor approximation, as proposed in Grathwohl et al. (2021); Zhang et al. (2022).
2. An objective to learn locally optimal proposals. Local optimality is defined as the proposal that minimizes the variance of the importance weights.

The authors show the benefits of their proposed gradient-based kernels and the learned proposal on image, text and DNA sequence design datasets, showing the efficacy of their approach over best of N and SMC without gradients or learned proposals.

**Strengths:**

The paper develops two different proposals for SMC for discrete diffusion models. Both proposals are rigorously motivated and are practical to implement, with the empirical results showing the efficacy of both approaches.

**Weaknesses:**

While the underlying results have been known (and cited) in the context of designing proposals for MCMC and discrete generative (Grathwohl et al. (2021); Zhang et al. (2022)). Their use in discrete diffusion is well motivated and shown to produce significant improvements.

**Questions:**

-

---

> ### Author Response · Authors · 2025-11-21
>
> Thanks for your comments and the favourable on our paper!
>
> > While the underlying results have been known (and cited) in the context of designing proposals for MCMC and discrete generative (Grathwohl et al. (2021); Zhang et al. (2022)). Their use in discrete diffusion is well motivated and shown to produce significant improvements.
> >
>
> **ANSWER:** We are very grateful for the reviewer’s positive feedback on our contribution. While proposal design has been extensively studied in the MCMC literature, our aim is to explore how these ideas can be adapted and applied to the SMC setting for discrete diffusion models. We hope that our study illustrates how thoughtfully designed proposal distributions can improve performance, and that it may encourage further investigation into proposal design for SMC in diffusion modelling.

---

### Official Review · Reviewer_rEgU · 2025-10-29

**Soundness:** 4
**Presentation:** 4
**Contribution:** 4
**Rating:** 8
**Confidence:** 4

**Summary:**

This paper studies how to perform inference-time control of a pretrained masked diffusion model (MDM), such as sampling from a reward-tilted distribution or sampling from "product of experts" (a more general notion of classifier-free guidance). The paper proposed an SMC framework, and discussed several design choices including how to choose the intermediate distributions, the proposal distributions, and light-weighted learning objectives compared to full fine-tuning.

**Strengths:**

The paper is well-written and clearly organized, with rigorous mathematical derivations, although some of the notations can be further improved for better clarity. The SMC framework for MDMs is novel and well-motivated. The experiments are also comprehensive, covering various tasks from synthetic toy examples to large scale language, DNA and image generation tasks, and the results demonstrate the effectiveness of the proposed methods. I appreciate the extensive ablation studies and analyses provide, and feel that the paper makes a solid contribution to the field of inference-time scaling of masked diffusion models.

**Weaknesses:**

I don't see any significant weakness, but one way this paper can be further improved is through including inference-time scaling baselines to compare with, which are not presented in the main text. I checked the appendix and found some comparisons with other fine-tuning methods such as DRAKES for the DNA task, but there is no comparison with purely inference-time control methods (e.g., SVDD, arxiv:2408.08252). The authors are encouraged to include such comparisons and report the results, but I understand that it may require significant additional effort and the current results are already quite strong and convincing.

**Questions:**

1. In section 3.3, the authors proposed to approximate $r(x _ t)$ by $\hat r(x _ t):=\mathbb{E} _ {p _ \theta(x _ 0|x _ t)}r(x _ 0)$, as the original $r(x _ t)$ may not be well-defined (e.g., $x _ t$ contains mask states, while $r$ is evaluated on clean sequences). Could the authors explain in more detail how to apply the Gumbel softmax trick in order to compute $\nabla _ {x _ t}\hat r(x _ t)$? It's supposed to be discussed in appendix C.2 but I couldn't find the details there. Also, here $p _ \theta(x _ 0|x _ t)$ is actually one-step denoising, assuming each masked position is sampled independently from the model output, right?

2. In proposition 2, the authors proposed to use the locally optimal proposal for variance reduction of the weights, and train an amortized proposal distribution $q _ \phi$ through the log-variance loss. Is it possible to visualize the effective sample size of the SMC with and without the learned proposal to demonstrate the performance gain?


Minor comments:

1. In line 101-107: it's better to add the superscript $\star^{(i)}$ to all $x _ t$ and $x _ {t-1}$'s to make it more clear and consistent.

2. In line 112-123: at first glance, I though that the aim is only to sample from the final target distributions $\pi(x _ 0)\propto p _ {\theta _ 1}^\alpha(x _ 0)p _ {\theta _ 2}^\beta(x _ 0)$ and $\pi(x _ 0)\propto p _ {\theta}(x _ 0)\exp(r(x _ 0))$, and the intermediate distributions are obtained by propagating these final targets through the same forward noising process as in the original MDM. But later I realized that these equations also specify *all* intermediate distributions $\pi(x _ t)$ along the way that you want your generated trajectories to follow. It would be helpful to clarify this point in the main text to avoid confusion.

3. In equation (7): $\exp(r(x _ 0))$ should be $\exp(r(x _ {t-1}))$?

4. In equation (9), the variance and summation should swap their positions? Actually we can directly write in the expectation over $t$ from uniform distribution on $1$ to $T$ so that there will be no $T^2$ term in proposition 3.

5. In proposition 1, the weight does not depend on the backward rate matrix $\hat R _ t$. I checked the proof and realized that this is because of the assumption on lines 1008-1009, which should be stated in the proposition.

---

> ### Author Response · Authors · 2025-11-21
>
> Thanks for your favour in our work! Below are our answers to your questions. Please feel free to leave additional comments if you have any further concerns or would like to discuss them further.
>
> > I don't see any significant weakness, but one way this paper can be further improved is through including inference-time scaling baselines to compare with, which are not presented in the main text. I checked the appendix and found some comparisons with other fine-tuning methods such as DRAKES for the DNA task, but there is no comparison with purely inference-time control methods (e.g., SVDD, arxiv:2408.08252). The authors are encouraged to include such comparisons and report the results, but I understand that it may require significant additional effort and the current results are already quite strong and convincing.
> >
>
> **ANSWER:**  We thank the reviewer for the reference to the baseline. There are two key differences between SVDD and our methods: i) SVDD applies importance sampling at each step, which is not equivalent to the SMC; ii) SVDD uses a pretrained diffusion model as the proposal, whereas we use an approximately optimal proposal. We compare our method with SVDD on both DNA sequence design and language modelling, with results shown in the following tables:
>
> DNA Sequence Design (see Table 5 in the paper for the comparisons to other baselines):
>
> | Method | Pred-Activity (median) ↑ | ATAC-Acc ↑ (%) | 3-mer Corr ↑ | JASPAR Corr ↑ | App-Log-Lik (median) ↑ |
> | --- | --- | --- | --- | --- | --- |
> | SVDD (N=8) | 6.57(0.01) | 67.4(0.01) | 0.813(0.009) | 0.753(0.011) | -258(0.2) |
> | SVDD (N=16) | 6.89(0.04) | 84.3(0.01) | 0.891(0.009) | 0.834(0.011) | -260(0.2) |
> | $\text{SMC}_{\text{amot}}$ (N=1) |  5.40(0.02) | 82.1(0.01) | 0.653(0.001) | 0.778(0.005) | -259(0.1) |
> | $\text{SMC}_{\text{amot}}$ (N=8) | 6.35(0.01) | 95.8(0.01) | 0.736(0.003) | 0.845(0.005) | -261(0.2) |
> | $\text{SMC}_{\text{amot}}$ (N=16) | 6.68(0.02) | 97.6(0.01) |  0.796(0.005) | 0.886(0.002) | -261(0.4) |
>
> The results show that, even with a single particle, our method performs on par with SVDD using multiple particles, highlighting the effectiveness of the proposed approximately optimal proposal. As the particle count increases, both SVDD and SMC continue to improve. However, we observe that while SVDD achieves higher predicted activity, it performs worse in ATAC accuracy. Note that the tilted reward used in this experiment is the predicted activity of the DNA sequence. This suggests that SVDD is somewhat prone to reward hacking, whereas SMC demonstrates better generalisation, as reflected by its stronger ATAC accuracy.
>
> Language Modelling (see Table 4 in the paper for the comparisons to other baselines):
>
> |  |  Toxic ↑  | Toxic (Holdout) ↑ | PPL (GPT2-XL) ↓ | Dist-1/2/3 ↑ |
> | --- | --- | --- | --- | --- |
> | SVDD (N=1) | 0.3% | 3.7% | 121.6 | 56/91/96 |
> | $\text{SMC}_{\text{amot}}$ (N=1) | 63.7% | 75.7% |  131.9 | 53/89/94 |
> | SVDD (N=2) | 14.6% | 27.0% | 129.0 | 56/91/95 |
> | $\text{SMC}_{\text{amot}}$ (N=2) | 84.7% |  90.3% | 140.6 |  51/88/94 |
> | SVDD (N=4) | 65.7% | 67.0% | 129.1 | 58/91/94 |
> | $\text{SMC}_{\text{amot}}$ (N=4) | 98.3% |  99.0% | 127.0 | 43/81/91 |
> | SVDD (N=8) | 92.2% | 82.2% | 121.9 | 59/90/93 |
> | $\text{SMC}_{\text{amot}}$ (N=8) | 100.0% |  100.0% | 127.0 | 43/81/91 |
> | SVDD (N=16) | 97.5% | 91.0% | 127.7 | 58/89/93 |
> | $\text{SMC}_{\text{amot}}$ (N=16) | 100.0% | 100.0% |  114.2 |  40/79/90 |
>
> The results show that our method, built on SMC and an approximately optimal proposal distribution, consistently achieves higher toxicity than SVDD, demonstrating the effectiveness of the proposed approach. Although the SMC-based methods sacrifice a small amount of perplexity, they still produce coherent text with elevated toxicity levels, as illustrated in Figure 16 in the paper.

---

> > ### Author Response · Authors · 2025-11-21
> >
> > > In section 3.3, the authors proposed to approximate $r(x_t)$ by $\hat{r}(x_{t}) := \mathbb{E}\_{p_{\theta} (x_{0} | x_{t} ) }[r(x_{0})]$, as the original $r(x_t)$ may not be well-defined (e.g., contains mask states, while is evaluated on clean sequences). Could the authors explain in more detail how to apply the Gumbel softmax trick in order to compute $\nabla_{x_{t}} \hat{r}(x_{t})$? It's supposed to be discussed in appendix C.2 but I couldn't find the details there. Also, here is actually one-step denoising, assuming each masked position is sampled independently from the model output, right?
> > >
> >
> > **ANSWER:** Thank you for pointing these out. You are right, the optimal tilted reward is $r(x_{t}) = \mathbb{E}\_{p_{\theta} (x_{0} | x_{t})} [r(x_{0})]$. In this paper, as shown in equation 12, we approximate this expectation via Monte Carlo estimation: $\hat{r}(x_{t}) = \frac{1}{M} \sum_{m=1}^M r(x_{0}^{m}), x_{0}^{m} \sim p_{\theta} (x_{0} | x_{t})$, where  $p_{\theta} (x_0 | x_t)$ is the one-step denoising model, and each masked position is sampled independently from its output.
> >
> > To compute $\nabla_{x_t} \hat{r}(x_t)$, we decompose the computation of $\hat{r}(x_t)$ into three steps:
> >
> > 1. compute the denoising logits: $p = \mu_\theta (x_t)$, where $\mu_\theta$ is the denoising model
> > 2. sample $x_0$: $x_0 \sim \mathrm{Cat}(x; p)$
> > 3. evaluate the reward $\hat{r}(x_t) = r(x_0)$
> >
> > For steps (1) and (3), following [1], we treat both $r$ and $\mu_{\theta}$ as functions that accept continuous real-valued inputs, so that their gradients are well defined. For step (2), we apply the reparameterization trick using the Gumbel–Softmax relaxation, which make $x_0$ differentiable with respect to $p$. Therefore,  $\nabla_{x_t} \hat{r}(x_t)$ can be computed via the chain rule as
> >
> > $$
> > \nabla_{x_t} \hat{r}(x_t) = \frac{\partial r(x_0)}{\partial x_0}\frac{\partial x_0}{\partial p}\frac{\partial p}{\partial x_t}
> > $$
> >
> > where the first and last terms arise from the differentiable relaxations of $r$ and $\mu_\theta$, and the middle term comes from the Gumbel–Softmax relaxation. We will include this discussion in Appendix C.2.
> >
> > [1] Grathwohl, Will, et al. "Oops i took a gradient: Scalable sampling for discrete distributions." *International Conference on Machine Learning*. PMLR, 2021.
> >
> > > In proposition 2, the authors proposed to use the locally optimal proposal for variance reduction of the weights, and train an amortized proposal distribution $q_{\phi}$ through the log-variance loss. Is it possible to visualize the effective sample size of the SMC with and without the learned proposal to demonstrate the performance gain?
> > >
> >
> > **ANSWER:** We thank the reviewer for the constructive suggestion. In response, we would like to visualise the effective sample size (ESS) for the MNIST synthetic experiment. Specifically, we apply three different proposals: $\text{Prop}\_{\text{pretrained}}$, $\text{Prop}\_{\text{grad}}$, $\text{Prop}\_{\text{amot}}$, corresponding to the pretrained diffusion proposal, the first-order approximated proposal, and the learned amortised proposal. We repeat each experiment using 16 particles over 30 independent runs, and then plot both the ESS and the reward trajectory across the sampling steps. It is important to note that we do not perform resampling in these experiments. Resampling would reset the importance weights at each step, thereby obscuring the natural evolution of the ESS and making it difficult to compare how effectively each proposal maintains particle diversity. By omitting resampling, the ESS provides a clearer reflection of the intrinsic quality of each proposal.
> >
> > We present the ESS and reward trajectories in [this figure](https://imgshare.cc/9di7h3ow), along with the generated [samples](https://imgshare.cc/4wkzjos9). The results show that $\text{Prop}\_{\text{amot}}$ achieves the highest ESS and reward throughout the sampling process, highlighting the effectiveness of the proposed log-variance minimisation objective in learning an approximately optimal proposal. In contrast, although $\text{Prop}\_{\text{grad}}$ attains higher reward values than $\text{Prop}\_{\text{pretrained}}$, it suffers from lower ESS. This is expected, as the first-order approximated proposal introduces bias relative to the true optimal proposal, making it more prone to reward hacking. Regarding the generated samples, $\text{Prop}\_{\text{grad}}$ fails to produce high-quality images, whereas $\text{Prop}\_{\text{amot}}$ consistently generates visually coherent and high-quality samples. This observation further confirms the overall superiority of the learned amortised proposal.

---

> > > ### Author Response · Authors · 2025-11-21
> > >
> > > > In line 112-123: at first glance, I though that the aim is only to sample from the final target distributions $\pi(x_0) \propto p_{\theta_1}^\alpha (x_0) p_{\theta_2}^\beta (x_0)$ and $\pi (x_0) \propto p_\theta (x_0) \exp(r(x_0))$, and the intermediate distributions are obtained by propagating these final targets through the same forward noising process as in the original MDM. But later I realized that these equations also specify *all* intermediate distributions along the way that you want your generated trajectories to follow. It would be helpful to clarify this point in the main text to avoid confusion.
> > > >
> > >
> > > **ANSWER:** Sorry for the confusion. Yes, we consider two settings $\pi (x_t) \propto p_{\theta_1}^\alpha (x_t) p_{\theta_2}^\beta (x_t)$ and $\pi (x_t) \propto p_\theta (x_t) \exp(r(x_t))$, where $p_{\theta_1}, p_{\theta_2}, p_{\theta}$ are all pretrained diffusion models. For each $t > 0$, these expressions define the intermediate target distributions$\pi(x_t)$ that we want the generated trajectories to follow. Setting $t=0$, recovers the final target distribution $\pi(x_0)$ at the endpoint of the trajectory. We will clarify this point in the main text to avoid potential confusion.
> > >
> > > > In equation (9), the variance and summation should swap their positions? Actually we can directly write in the expectation over $t$ from uniform distribution on $1$ to $T$ so that there will be no $T^2$ term in proposition 3.
> > > >
> > >
> > > **ANSWER:** Thank you for the insightful comment. Your intuition is absolutely reasonable. In practice, one can indeed sample the time step $t$ uniformly from $\{0,\dots,T\}$, evaluate the loss at that step, and update the model accordingly. This procedure avoids the explicit $T^2$ factor and corresponds to how the objective is implemented. However, this surrogate loss serves as an *upper bound* on the true log-variance objective and is not an equivalent reformulation.
> > >
> > > To make this precise, recall that in Equation (5) the importance weights evolve recursively:
> > >
> > > $$
> > > w_t = \frac{\exp(r(x_{t}))}{\exp(r(x_{t+1}))}\frac{p_\theta (x_{t} | x_{t+1})}{q_\phi (x_{t} | x_{t+1})} w_{t+1}
> > > $$
> > >
> > > Thus, the log-weight at $t$ is
> > >
> > > $$
> > > w_t = \sum_{s = t}^T \log \frac{\exp(r(x_{s}))}{\exp(r(x_{s+1}))}\frac{p_\theta (x_{s} | x_{s+1})}{q_\phi (x_{s} | x_{s+1})}
> > > $$
> > >
> > > In this trajectory-level importance sampling view (i.e., without resampling), the importance weights accumulate multiplicatively across all future steps. Consequently, the variance must be taken after summing these incremental contributions:
> > >
> > > $$
> > > \mathbb{V}\_{q} [\log w_t] = \mathbb{V}\_{q} [\sum_{s = t}^T \log \frac{\exp(r(x_{s}))}{\exp(r(x_{s+1}))}\frac{p_{\theta} (x_{s} | x_{s+1})}{q_\phi (x_{s} | x_{s+1})}]
> > > $$
> > >
> > > which corresponds exactly to the form of Equation (9) in the paper.
> > >
> > > In contrast, much of the classical SMC literature assumes resampling at every step (or at least analyses the fully resampled case). Under resampling, the weight is reset so that $w_{t+1}=1$, and the incremental weight becomes
> > >
> > > $$
> > > w_t = \frac{\exp(r(x_{t}))}{\exp(r(x_{t+1}))}\frac{p_\theta (x_{t} | x_{t+1})}{q_\phi (x_{t} | x_{t+1})}
> > > $$
> > >
> > > Here, the log-weights do not accumulate, and the variance acts only on the local term. In this special case, the variance and the summation can be interchanged.
> > >
> > > To remain fully rigorous in the setting without per-step resampling, we place the summation inside the variance operator, which yields Equation (9) as presented.
> > >
> > > > In equation (7): should $\exp(r(x_0))$ be $\exp(r(x_{t-1}))$?
> > > >
> > >
> > > **ANSWER:** Thank you for pointing this out. We checked Equation (7), and we do not find a term of the form $\exp(r(x_0))$ in equation 7. The original equation appears to be correct as written.
> > >
> > > > In line 101-107: it's better to add the superscript to all and 's to make it more clear and consistent.
> > > In proposition 1, the weight does not depend on the backward rate matrix $\hat{R}_t$. I checked the proof and realized that this is because of the assumption on lines 1008-1009, which should be stated in the proposition.
> > > >
> > >
> > > **ANSWER:** Thanks for pointing these out. We will revise the paper accordingly.

---

> > > > ### Comment · Reviewer_rEgU · 2025-11-24
> > > >
> > > > I thank the authors for the rebuttal, which has fully addressed all of my concerns. The comparison with SVDD and the ESS results are quite nice. The explanation of the Gumbel softmax trick looks clear for me, and I hope in your revised paper more details for doing this differentiable relaxation can be included. Regarding question 3, sorry that I made a typo: I mean the $\exp(r(x _ s))$ on the left hand side, which should be $\exp(r(x _ {t-1}))$.
> > > >
> > > > Please don't forget to include all changes in the revision. ICLR allows the authors to modify the paper during rebuttal, so it's better to update your paper now. Overall, I maintain a positive view of this paper and strongly recommend it for acceptance.

---

> > > > > ### Author Response · Authors · 2025-11-24
> > > > >
> > > > > Thank you for your positive feedback on our work! We apologise for the typo. You are absolutely right. In Equation 7, $\exp(r(x_{s}))$ should be $\exp(r(x_{t-1}))$. We are currently revising the paper and will upload the updated version soon. Thank you again for your constructive feedback. It truly helps us improve our work.

---

### Official Review · Reviewer_MEfE · 2025-11-03

**Soundness:** 2
**Presentation:** 3
**Contribution:** 2
**Rating:** 2
**Confidence:** 5

**Summary:**

Based on the motivation from existing work on the inference-time scaling framework of continuous diffusion models, this paper also used the Sequential Monte Carlo (SMC) method to develop an inference-time scaling/control framework of pretrained discrete diffusion models, with a particular focus on reward-tilted sampling (posterior sampling) and classifier-free guidance (CFG). Specifically, tractable importance weights are used throughout the inference process to sample from either reward-tilted or product-of-experts targets with pretrained discrete diffusion models. To avoid possible problems incurred by particle degeneracy, the authors also proposed a way to approximate that the locally optimal proposal that minimizes the variance of the particles' weights. Numerical experiments on synthetic examples, language modeling, biology design and text-to-image generation are provided to justify the effectiveness of the proposed method.

**Strengths:**

The reviewer finds that the paper is clearly written and explained.

**Weaknesses:**

The reviewer thinks that this paper does have quite a few weaknesses, which are listed below:

(1) Firstly, it seems that the novelty of the SMC-based inference-time scaling framework seems to be quite limited. Specifically, the distribution path (with respect to time time variable $t$) is the same as that of [1], even though the authors did provide a way to justify how the discrete-time formulation proposed in this paper converges to [1] under the continuum limit. Moreover, the idea of variance reduction seems to be the same as that of [2]. In addition, to the best of the reviewer's knowledge, the inference-time scaling of continuous and discrete diffusion models have emerged as a quite hot topic these days, but the current version of the manuscript's literature review section is missing a bunch of related work [3,4,5,6,7,8,9,10,11,12,13,14,20,21] (including works studying the problem of fine-tuning continuous/discrete diffusion models), especially the ones [3,4,6,7,20,21] that developed SMC-based methods for sampling the reward-tilted or product-of-experts targets. Specifically for the setting of discrete diffusion models, the reviewer also note that there exists a few concurrent work [15,16] studying the same topic as this paper. Overall, the quality and positioning of this paper can be possibly improved by citing all work (related and concurrent) above and discussing them appropriately.

(2) Secondly, the reviewer is a bit concerned that there are not enough baselines included in the current version of this paper. In particular, the authors only included 3 methods that are not SMC-based as baselines for large-scale experiments like language modeling, biology design and text-to-image generation. To the best of the reviewer's knowledge, these methods are not state-of-the-art (SOTA) within the class of methods that use only a single particle. For instance, one other more effective methods that the authors might consider including as baselines is the one [8,17] based on Gibbs sampling.

(3) Thirdly, the reviewer finds the theory part presented in the paper to be a bit weak. Just as what has been done in the theoretical part of previous work [18,19] that develops SMC-based methods for diffusion-based sampling, the authors should consider studying what is the final error between the targe distribution (reward-tilted or product-of-experts targets) and the distribution returned by the SMC algorithm by adopting appropriate assumptions (for instance, score matching error).

**Questions:**

Following (3) discussed in the weaknesses section above, would it be possible for the authors to provide some intuition on how a theoretical analysis for the discrete-time formulation proposed in this paper? Specifically, what might be the analogies of the assumptions and theoretical results in previous work [18,19] for the SMC-based inference-time scaling of discrete diffusion models?

Furthermore, it seems to the reviewer that the approach proposed in subsection 3.2.1 needs to have access to the gradient of the reward function $r$, but sometimes $r$ might not be differentiable. Would it be possible for the authors to come up with a modified version of equation (8) to approximate the optimal proposal in this case?

Conclusively speaking, the reviewer thinks that this paper is probably not ready for a top ML venue like ICLR and needs further revision.

References:

[1] Lee, Cheuk Kit, Paul Jeha, Jes Frellsen, Pietro Lio, Michael Samuel Albergo, and Francisco Vargas. "Debiasing guidance for discrete diffusion with sequential monte carlo." arXiv preprint arXiv:2502.06079 (2025).

[2] Holderrieth, Peter, Michael S. Albergo, and Tommi Jaakkola. "LEAPS: A discrete neural sampler via locally equivariant networks." arXiv preprint arXiv:2502.10843 (2025).

[3] Trippe, B. L., Yim, J., Tischer, D., Baker, D., Broderick, T., Barzilay, R., & Jaakkola, T. (2022). Diffusion probabilistic modeling of protein backbones in 3d for the motif-scaffolding problem. In The Eleventh International Conference on Learning Representations (ICLR), 2023.

[4] Dou, Z., & Song, Y. (2024). Diffusion posterior sampling for linear inverse problem solving: A filtering perspective. In The Twelfth International Conference on Learning Representations (ICLR), 2024.

[5] Ma, Nanye, Shangyuan Tong, Haolin Jia, Hexiang Hu, Yu-Chuan Su, Mingda Zhang, Xuan Yang et al. "Scaling Inference Time Compute for Diffusion Models." In Proceedings of the Computer Vision and Pattern Recognition Conference, pp. 2523-2534. 2025.

[6] Chen, Haoxuan, Yinuo Ren, Martin Renqiang Min, Lexing Ying, and Zachary Izzo. "Solving inverse problems via diffusion-based priors: An approximation-free ensemble sampling approach." arXiv preprint arXiv:2506.03979 (2025).

[7] Yoon, Taehoon, Yunhong Min, Kyeongmin Yeo, and Minhyuk Sung. "$\Psi $-Sampler: Initial Particle Sampling for SMC-Based Inference-Time Reward Alignment in Score Models." arXiv preprint arXiv:2506.01320 (2025).

[8] Dang, Meihua, Jiaqi Han, Minkai Xu, Kai Xu, Akash Srivastava, and Stefano Ermon. "Inference-time scaling of diffusion language models with particle gibbs sampling." arXiv preprint arXiv:2507.08390 (2025).

[9] Chen, Tianlang, Minkai Xu, Jure Leskovec, and Stefano Ermon. "RFG: Test-Time Scaling for Diffusion Large Language Model Reasoning with Reward-Free Guidance." arXiv preprint arXiv:2509.25604 (2025).

[10] Ren, Yinuo, Wenhao Gao, Lexing Ying, Grant M. Rotskoff, and Jiequn Han. "Driftlite: Lightweight drift control for inference-time scaling of diffusion models." arXiv preprint arXiv:2509.21655 (2025).

[11] Zhang, Xiangcheng, Haowei Lin, Haotian Ye, James Zou, Jianzhu Ma, Yitao Liang, and Yilun Du. "Inference-time scaling of diffusion models through classical search." arXiv preprint arXiv:2505.23614 (2025).

[12] Jain, Vineet, Kusha Sareen, Mohammad Pedramfar, and Siamak Ravanbakhsh. "Diffusion Tree Sampling: Scalable inference-time alignment of diffusion models." arXiv preprint arXiv:2506.20701 (2025).

[13] Ramesh, Vignav, and Morteza Mardani. "Test-Time Scaling of Diffusion Models via Noise Trajectory Search." arXiv preprint arXiv:2506.03164 (2025).

[14] Tang, Sophia, Yuchen Zhu, Molei Tao, and Pranam Chatterjee. "TR2-D2: Tree Search Guided Trajectory-Aware Fine-Tuning for Discrete Diffusion." arXiv preprint arXiv:2509.25171 (2025).

[15] Hasan, Mohsin, Marta Skreta, Alan Aspuru-Guzik, Yoshua Bengio, and Kirill Neklyudov. "Discrete feynman-kac correctors." In 2nd AI for Math Workshop@ ICML 2025. 2025.

[16] Rout, Litu, Andreas Lugmayr, Yasamin Jafarian, Srivatsan Varadharajan, Constantine Caramanis, Sanjay Shakkottai, and Ira Kemelmacher-Shlizerman. "Test-Time Anchoring for Discrete Diffusion Posterior Sampling." arXiv preprint arXiv:2510.02291 (2025).

[17] Chu, Wenda, Zihui Wu, Yifan Chen, Yang Song, and Yisong Yue. "Split gibbs discrete diffusion posterior sampling." arXiv preprint arXiv:2503.01161 (2025).

[18] Wu, Luhuan, Brian Trippe, Christian Naesseth, David Blei, and John P. Cunningham. "Practical and asymptotically exact conditional sampling in diffusion models." Advances in Neural Information Processing Systems 36 (2023): 31372-31403.

[19] Cardoso, Gabriel, Yazid Janati El Idrissi, Sylvain Le Corff, and Eric Moulines. "Monte Carlo guided diffusion for Bayesian linear inverse problems." arXiv preprint arXiv:2308.07983 (2023).

[20] He, Jiajun, José Miguel Hernández-Lobato, Yuanqi Du, and Francisco Vargas. "RNE: a plug-and-play framework for diffusion density estimation and inference-time control." arXiv preprint arXiv:2506.05668 (2025).

[21] He, Jiajun, Paul Jeha, Peter Potaptchik, Leo Zhang, José Miguel Hernández-Lobato, Yuanqi Du, Saifuddin Syed, and Francisco Vargas. "CREPE: Controlling Diffusion with Replica Exchange." arXiv preprint arXiv:2509.23265 (2025).

---

> ### Author Response · Authors · 2025-11-21
>
> We appreciate the reviewer's constructive feedback. Regarding the weaknesses pointed out, we would like to provide the following clarifications:
>
> > Firstly, it seems that the novelty of the SMC-based inference-time scaling framework seems to be quite limited. Specifically, the distribution path (with respect to the time variable $t$) is the same as that of [1], even though the authors did provide a way to justify how the discrete-time formulation proposed in this paper converges to [1] under the continuum limit. Moreover, the idea of variance reduction seems to be the same as that of [2].
> >
>
> **ANSWER:** We thank the reviewer for raising the concerns regarding the novelty. To address your concern, we would like to reiterate our main contribution as:
>
> 1. An SMC framework tailored for discrete diffusion models.
>
>     While SMC has been widely applied in continuous diffusion settings (e.g., sampling from Boltzmann distributions and solving inverse problems), these methods rely on properties of continuous spaces and cannot be directly transferred to discrete domains. Discrete diffusion involves fundamentally different noise processes and transition kernels, leading to challenges not present in the continuous case. Our work provides a principled SMC formulation for discrete diffusion, including (i) an approximate yet stable computation of importance weights for mask-based diffusions and (ii) effective approaches to approximate the optimal proposals. These components are not inherited from continuous-diffusion SMC methods and are crucial for enabling inference-time scaling in discrete diffusion models.
>
> 2. A comprehensive study of the proposal design in SMC.
>
>     Although SMC has been used in continuous diffusion models, the design of effective proposal distributions, which is crucial to the performance and stability of SMC, remains relatively underexplored. In this work, we analyse the structure of the optimal proposal in discrete diffusion and introduce several principled approximations that make it computationally tractable. Our experiments further demonstrate that improved proposal choices lead to substantial gains in SMC effectiveness. We hope these observations provide insights that will inspire more rigorous proposal design in future research, ultimately advancing the performance of SMC in diffusion models.
>
>
> Regarding the related works [1, 2], we would like to distinguish our contribution from theirs:
>
> - [1] considers conditional generation. Although it can be seen as a special case of reward-tilted targets, [1] mainly focuses on continuous time and uses pretrained diffusion models as proposals, which corresponds exactly to one of our variant methods $\text{SMC}_{\text{base}}$. In Proposition 4 (Appendix B.2), we further show that our discrete-time SMC formulation converges to the continuous-time SMC in [1].
>
>     Moreover, we propose several methods to approximate the optimal proposal and demonstrate that better approximations lead to consistently improved SMC performance. This insight is not apparent from [1], which does not explore proposal design beyond using the pretrained model.
>
> - As noted in the remark in Appendix B.2, [2] considers different intermediate target distributions, defined via a geometric interpolation between the base and target distributions:  $\pi (x_t) \propto p_{\text{base}}^t (x_t) p_{\text{target}}^{1-t} (x_t)$ . In contrast, our work focuses on reward-tilted and product target distributions, both of which are intractable even up to a normalising constant. This distinction leads to substantially different challenges when applying SMC.
>
>     We also note that both [2] and our work employ log-variance minimisation to train proposal distributions. As cited in our paper, this objective has been widely used in previous work [Richter et al] on training continuous neural samplers and motivates both approaches.
>
>     Furthermore, [2] evaluates only on synthetic data such as Ising models, whereas we consider significantly larger-scale and more realistic settings. We demonstrate the effectiveness of discrete SMC across a diverse range of real-world tasks, including biological sequence design, language modelling, and text-to-image generation.
>
>
> We hope this clarification helps address the reviewer’s concerns regarding novelty. We believe that the exploration of the proposal design of SMC, along with the broader applications and evaluations, may offer useful insights and contribute meaningfully to the ongoing research in this area.
>
> Reference:
>
> Richter, Lorenz, et al. "Vargrad: a low-variance gradient estimator for variational inference." *Advances in Neural Information Processing Systems* 33 (2020): 13481-13492.
>
> Richter, Lorenz, and Julius Berner. "Improved sampling via learned diffusions." *arXiv preprint arXiv:2307.01198* (2023).

---

> ### Author Response · Authors · 2025-11-21
>
> > In addition, to the best of the reviewer's knowledge, the inference-time scaling of continuous and discrete diffusion models have emerged as a quite hot topic these days, but the current version of the manuscript's literature review section is missing a bunch of related work [3,4,5,6,7,8,9,10,11,12,13,14,20,21] (including works studying the problem of fine-tuning continuous/discrete diffusion models), especially the ones [3,4,6,7,20,21] that developed SMC-based methods for sampling the reward-tilted or product-of-experts targets. Specifically for the setting of discrete diffusion models, the reviewer also note that there exists a few concurrent work [15,16] studying the same topic as this paper. Overall, the quality and positioning of this paper can be possibly improved by citing all work (related and concurrent) above and discussing them appropriately.
> >
>
> **ANSWER:** Thank you for the references. They are highly relevant, and we will incorporate them into the revision. To better position their relationships to our work, we provide a summary table that organizes these works along five dimensions: (i) whether the method is SMC-based; (ii) whether it targets continuous or discrete diffusion models; (iii) whether it is concurrent with our submission (following the ICLR policy that papers appearing on or after July 24, 2025 are considered concurrent); (iv) whether it is already cited in our paper; and (v) whether it is published. As the table shows, none of the referenced works simultaneously meet all of the following criteria: SMC-based, focused on discrete diffusion models, non-concurrent, not yet cited, and published.
>
> Regarding works [3,4,6,7,20,21], although they also address related problems, our work differs in two essential ways:
>
> - These works apply SMC to continuous diffusion models, whereas our focus is on the *discrete* diffusion setting.
> - More importantly, beyond directly applying SMC, we investigate the design of proposal distributions, a central but under-explored component of SMC when used for diffusion models, showing that careful proposal construction yields substantial gains.
>
> Works [15,16] are also highly relevant, as both apply SMC to reward-tilted targets in discrete diffusion. However, they are not training-based methods and do not explore proposal design. Additionally, we were not aware of these papers at submission time: [15] is a workshop paper that appeared after July and was not posted on arXiv, and [16] is a new arXiv paper (posted on Oct 2) after our ICLR submission.
>
> Nevertheless, all these works are closely related to our paper and together provide a useful overview of ongoing research in test-time scaling for diffusion models. We will incorporate an appropriate discussion of them in the revision.
>
> |  | SMC-based Method | Diffusion Type (Continuous / Discrete) | Concurrent w.r.t. Our Submission (Date Online) | Already Cited in Our Paper | Published |
> | --- | --- | --- | --- | --- | --- |
> | Expected (ideal missing category) | ✔️ | Discrete | ❌ | ❌ | ✔️ |
> | [1] | ✔️ | Discrete | ❌ | ✔️ | ✔️ |
> | [2] | ✔️ | Discrete | ❌ | ✔️ | ✔️ |
> | [3] | ✔️ | Continuous | ❌ | ❌ | ✔️ |
> | [4] | ✔️ | Continuous | ❌ | ❌ | ✔️ |
> | [5] | ❌ | Continuous | ❌ | ❌ | ✔️ |
> | [6] | ✔️ | Continuous | ❌ | ❌ | ❌ |
> | [7] | ✔️ | Continuous | ❌ | ❌ | ✔️ |
> | [8] | ✔️ | Discrete | ❌ | ✔️ | ❌ |
> | [9] | ❌ | Discrete | ✔️ (29 Sept) | ❌ | ❌ |
> | [10] | ✔️ | Continuous | ✔️ (25 Sept) | ❌ | ❌ |
> | [11] | ❌ | Continuous | ❌ | ❌ | ❌ |
> | [12] | ❌ | Continuous | ❌ | ❌ | ✔️ |
> | [13] | ❌ | Continuous | ❌ | ❌ | ✔️ |
> | [14] | ❌ | Discrete | ✔️ (29 Sept) | ❌ | ❌ |
> | [15] | ✔️ | Discrete | ❌ | ❌ | ❌ |
> | [16] | ❌ | Discrete | ✔️ (2 Oct) | ❌ | ❌ |
> | [17] | ❌ | Discrete | ❌ | ❌ | ✔️ |
> | [18] | ✔️ | Continuous | ❌ | ✔️ | ✔️ |
> | [19] | ✔️ | Continuous | ❌ | ✔️ | ✔️ |
> | [20] | ✔️ | Continuous | ❌ | ✔️ | ❌ |
> | [21] | ❌ | Continuous | ✔️ (27 Sept) | ❌ | ❌ |

---

> > ### Author Response · Authors · 2025-11-21
> >
> > > Secondly, the reviewer is a bit concerned that there are not enough baselines included in the current version of this paper. In particular, the authors only included 3 methods that are not SMC-based as baselines for large-scale experiments like language modeling, biology design and text-to-image generation. To the best of the reviewer's knowledge, these methods are not state-of-the-art (SOTA) within the class of methods that use only a single particle. For instance, one other more effective methods that the authors might consider including as baselines is the one [8,17] based on Gibbs sampling.
> > >
> >
> > **ANSWER:** We thank the reviewer for pointing us to these baselines. We find both [8] and [17] highly relevant, and we would first like to clarify the main differences between our work and theirs:
> >
> > - While [8] also investigates SMC for discrete diffusion models, it is largely complementary to our work. In general, SMC methods involve three key components: (i) the computation of importance weights, (ii) the design of proposal distributions, and (iii) the choice of the SMC algorithm. Our work focuses primarily on the first two components, whereas [8] focuses on the third. In other words, although [8] proposes a Particle Gibbs SMC algorithm using a pretrained diffusion model as the proposal, our approximately optimal proposal can be integrated into their framework to further improve performance. Conversely, our method can also benefit from alternative SMC algorithms such as Particle Gibbs with Parallel Tempering [Puri et al.]. We leave this investigation for future work.
> > - [17] is a great baseline. However, as noted in the Limitations section (Appendix E.1) of [17], their method is only applicable to uniform noising processes. It remains unclear how their approach could be extended to other types of discrete diffusion models, such as masked diffusion models. In contrast, although our work primarily focuses on masked diffusion models, our method is applicable to both uniform and masked noising processes.
> >
> > Nevertheless, we include a comparison between our method and [17]. We do not compare against [8] because it is a complementary line of work, remains unpublished, and does not provide open-source code for reproducibility. The tables below present the results on biological design (see Table 5 in the paper for the comparisons to other baselines).
> >
> > | Method | Pred-Activity (median) ↑ | ATAC-Acc ↑ (%) | 3-mer Corr ↑ | JASPAR Corr ↑ | App-Log-Lik (median) ↑ |
> > | --- | --- | --- | --- | --- | --- |
> > | SGDD ($\beta = 30$) | 8.85(0.07) | 90.9(0.00) | 0.470(0.014) | 0.466(0.015) | -263(1.6) |
> > | SGDD ($\beta = 50$) | 9.32(0.04) | 96.4(0.01) | 0.370(0.010) | 0.398(0.001) | -269(0.1) |
> > | $\text{SMC}_{\text{amot}}$ (N=1) |  5.40(0.02) | 82.1(0.01) | 0.653(0.001) | 0.778(0.005) | -259(0.1) |
> > | $\text{SMC}_{\text{amot}}$ (N=8) | 6.35(0.01) | 95.8(0.01) | 0.736(0.003) | 0.845(0.005) | -261(0.2) |
> > | $\text{SMC}_{\text{amot}}$ (N=16) | 6.68(0.02) | 97.6(0.01) |  0.796(0.005) | 0.886(0.002) | -261(0.4) |
> >
> > This shows that although SGDD achieves higher predicted activity and ATAC accuracy, it performs much worse in terms of correlation. This suggests that SGDD may be overfitting to a single mode and is unable to generate diverse samples. In contrast, our SMC methods produce samples with both high activity and accuracy while maintaining diversity.
> >
> > Reference:
> >
> > Puri, Isha, et al. "A probabilistic inference approach to inference-time scaling of llms using particle-based monte carlo methods." *arXiv preprint arXiv:2502.01618* (2025).

---

> ### Author Response · Authors · 2025-11-21
>
> > Thirdly, the reviewer finds the theory part presented in the paper to be a bit weak. Just as what has been done in the theoretical part of previous work [18,19] that develops SMC-based methods for diffusion-based sampling, the authors should consider studying what is the final error between the targe distribution (reward-tilted or product-of-experts targets) and the distribution returned by the SMC algorithm by adopting appropriate assumptions (for instance, score matching error).
> >
>
> **ANSWER:**  Thank you for raising this concern. We believe there may be a slight misunderstanding regarding our setting. Using the reward-tilted case as an example, the target distribution is $\pi(x) \propto p_{\theta}(x) \exp(r(x))$, where $p_\theta$ is the pretrained model distribution. In other words, the objective is to tilt the pretrained model, not to the original data distribution used during pretraining. As a result, analysis based on score-matching error from the pretraining phase does not directly apply here.
>
> We also note that previous work [18,19] considers the same setting (i.e., tilting on the pretrained model rather than the original data distribution), and establishes asymptotic convergence guarantees for SMC in the continuous case without considering score matching error. Although our work focuses on discrete diffusion models, the same style of convergence analysis continues to apply in our setting (see the detailed answer below).
>
> > Following (3) discussed in the weaknesses section above, would it be possible for the authors to provide some intuition on how a theoretical analysis for the discrete-time formulation proposed in this paper? Specifically, what might be the analogies of the assumptions and theoretical results in previous work [18,19] for the SMC-based inference-time scaling of discrete diffusion models?
> >
>
> **ANSWER:** Our method is a direct application of classical Sequential Monte Carlo to the path measure induced by the intermediate target distributions $\{\pi_t\}_{t=0}^{T}$. The relevant theoretical guarantees therefore follow from standard SMC theory. In particular, under the usual assumptions that
>
> - the proposal kernels $q_\phi (x_{t} | x_{t+1})$ place positive mass on the support of $\pi_t$;
> - the incremental importance weights admit bounded moments across $t$;
> - the forward Markov kernels, which in our case correspond to the diffusion model’s fixed noising transitions, satisfy mixing conditions;
>
> well-established results [Freitas et al, Moral et al, Chopin et al] of SMC guarantee that the particle system consistently approximates each $\pi_t$. In particular, for any test function $\psi$, the particle estimator $N^{-1} \sum\_{i=1}^N \psi(x_t^{(i)})$ converges to $\mathbb{E}\_{\pi(x_{t})}[\psi(x_t)]$, obeys a law of large numbers, and admits a central limit theorem with asymptotic variance of $\mathcal{O}(1/N)$. Since the forward diffusion kernels naturally satisfy strong mixing properties and the learned proposal helps control weight variance, these assumptions carry over directly to our discrete-diffusion setting, allowing the classical SMC convergence guarantees to apply without modification.
>
> Reference:
>
> De Freitas, Nando. *Sequential Monte Carlo methods in practice*. Eds. Arnaud Doucet, and Neil James Gordon. Vol. 1. No. 2. New York: springer, 2001.
>
> Moral, Pierre. *Feynman-Kac formulae: genealogical and interacting particle systems with applications*. Springer New York, 2004.
>
> Chopin, Nicolas. "Central limit theorem for sequential Monte Carlo methods and its application to Bayesian inference." (2004): 2385-2411.
>
> > Furthermore, it seems to the reviewer that the approach proposed in subsection 3.2.1 needs to have access to the gradient of the reward function , but sometimes might not be differentiable. Would it be possible for the authors to come up with a modified version of equation (8) to approximate the optimal proposal in this case?
> >
>
> **ANSWER:** Thank you for raising this question. When the reward function is non-differentiable, the first-order approximation used in Equation (8) is indeed not applicable. In such cases, one can instead draw on approximation techniques developed in the discrete MCMC literature. For example, previous work [Yue et al] proposes using Newton’s series approximation (equivalently, the multilinear extension) to approximate the optimal proposal without requiring gradients of the reward. This provides a practical alternative when differentiability is not available.
>
> Reference:
>
> Xiang Yue, et al. "Efficient informed proposals for discrete distributions via newton’s series approximation." *International Conference on Artificial Intelligence and Statistics*. PMLR, 2023.

---

### Author Response · Authors · 2025-11-24
**Summary (1/2)**

We thank all reviewers for their constructive comments. First, we would like to summarise the paper according to the reviewers:

- Our paper introduces an effective test-time scaling framework for discrete diffusion models using Sequential Monte Carlo. We systematically study SMC proposal design and show that more nearly optimal proposals yield stronger performance. Reviewers agree with the technical soundness of our approach, and consider our paper to be well-motivated and well-written.

    > R**MEfE**: “The reviewer finds that the paper is clearly written and explained.”
    R**rEgU**: “The paper is well-written and clearly organized, with rigorous mathematical derivations; The SMC framework for MDMs is novel and well-motivated.”
    R**vdGL**: “The paper develops two different proposals for SMC for discrete diffusion models. Both proposals are rigorously motivated and are practical to implement.”
    >
- We demonstrate the versatility of the proposed discrete sampler across a broad range of applications, including synthetic experiments, biology design, language modelling, and text-to-image generation. The reviewers generally consider our paper to be well-supported by experiments.

    > R**rEgU**: “The experiments are also comprehensive; I appreciate the extensive ablation studies and analyses provide, and feel that the paper makes a solid contribution to the field of inference-time scaling of masked diffusion models”
    R**vdGL**: “the empirical results showing the efficacy of both approaches.”
    R**jzG5**: “The paper is proposes a new fine-tuning method for discrete diffusion models and demonstrate its performance by numerical examples.”
    >

---

> ### Author Response · Authors · 2025-11-24
> **Summary (2/2)**
>
> We now summarise our responses to the main concerns here, with detailed replies provided for each reviewer below. We would be glad to answer any further questions the reviewer may have after reviewing the answers:
>
> 1. The novelty of the proposed method
>
>     > The reviewers **MEfE** raise concerns about the nolvety, expecically considering  inference-time scaling have emerged as a quite hot topic these day and our manucripts is missing some related works.
>     >
>
>     We appreciate the reviewer’s comprehensive list of references provided. We have included them into the revision. To address the reviewer’s concern, we provide a brief overview of our contributions here, with a more detailed discussion included in our [response](https://openreview.net/forum?id=7wbrFQvfdH&noteId=eah248T5gV) to reviewer **MEfE.**
>
>     - An SMC framework specifically designed for discrete diffusion models.
>
>         While SMC methods have been successfully applied in continuous diffusion, such as Boltzmann sampling and inverse problem solving, these techniques depend heavily on properties of continuous spaces and cannot be directly transferred to discrete domains. Discrete diffusion, with its distinct noise process and transition structures, poses unique challenges that do not arise in the continuous setting. In response, we develop a principled SMC framework specifically for discrete diffusion, featuring (i) a stable yet tractable approach to computing importance weights for mask-based diffusions, and (ii) practical approximations for the optimal proposal distribution. These contributions extend existing continuous-diffusion SMC approaches and are critical for achieving scalable inference in discrete diffusion models.
>
>     - A systematic study of proposal design for SMC in discrete diffusion.
>
>         Although SMC has appeared in previous work on continuous diffusion, the question of how to construct effective proposals, arguably the key factor governing stability and efficiency, has received limited attention. In the discrete setting, we revisit this issue from first principles, characterising the behaviour of the optimal proposal and developing practical approximations that retain its essential structure while remaining tractable. Empirically, these choices lead to clear performance gains, underscoring the importance of proposal quality. We believe these observations can help guide subsequent efforts toward more deliberate and practical proposal design for diffusion models.
>
> 2. Comparison to other baselines
>
>     > Reviewers **MEfE** and **rEgU** are encouraged us to compare our methods to more baselines.
>     >
>
>     We have added SVDD and SGDD as new baselines. As shown in our [response](https://openreview.net/forum?id=7wbrFQvfdH&noteId=u7lWqZKGUR) to r**MEfE**,  SGDD exhibits mode collapse and limited sample diversity. In contrast, our SMC methods achieve strong activity and accuracy while preserving diversity. Moreover, SVDD appears to suffer from reward-hacking, whereas SMC showcases better generalisation (see our response to r**rEgU** for details).
>
> 3. Tracing the effective sample size
>
>     > Reviewers **rEgU** and **jzG5** are curious about the effective sample size of the learned amortised optimial proposal, as it serves as an indicator of sample diversity.
>     >
>
>     We appreciate the reviewers’ very constructive suggestions. In response, we conduct experiments on MNIST synthetic tasks. As shown in our [response](https://openreview.net/forum?id=7wbrFQvfdH&noteId=cLmoWWiIrT) to r**jzG5** and the [response](https://openreview.net/forum?id=7wbrFQvfdH&noteId=gHtRBzlg4P) to r**rEgU**, $\text{Prop}\_{\text{amot}}$ achieves the highest ESS and reward across the sampling process, demonstrating the effectiveness of the proposed log-variance minimisation objective in learning an approximately optimal proposal. Although $\text{Prop}\_{\text{grad}}$ attains higher rewards than $\text{Prop}\_{\text{pretrained}}$, it shows lower ESS. This is expected, since the first-order approximation introduces bias relative to the true optimal proposal, making it more susceptible to reward hacking.

---

### Author Response · Authors · 2025-11-24
**Summary of the Revision**

We sincerely thank all reviewers for their time and thoughtful feedback provided, as it has significantly contributed to improving our work. Now the revised version is ready. We summarise the main revisions in the following:

- As the reviewer **MEfE** and **rEgU** suggested, we have included new baselines for language modelling and biology design in Appendix D.2.3 and D.2.4.
- As the reviewer **rEgU** and **jzG5** suggested, we have included a study of the effective sample size along the sampling trajectory in Appendix D.2.2
- As the reviewer **rEgU** suggested, we have included a detailed discussion of the Gumbel-Softmax trick in Appendix C.2.
- As the reviewer **jzG5** suggested, we have included a brief recap of SMC in Appendix A.1.
- As the reviewer **MEfE** suggested, we have included the reference you mentioned in Section 5.

Once again, we sincerely appreciate the constructive feedback you’ve provided. We would be glad to hear any further suggestions you might have that could help improve our work.

---

### Meta-Review · Area_Chair_65S8 · 2026-01-06

**Summary:**

This paper is on the inference-time scaling of discrete diffusion models. The authors propose a method based on the well known sequential Monte Carlo (SMC) framework. Several technical modifications are provided to make the method tractable for discrete diffusion models. The method is exemplified on two types of target distributions: product target and reward-tilting distribution. Experiments are designed to demonstrate the advantages of the proposed algorithm.

**Reviewer Concerns:**

Most reviewers think the paper is interesting and solid. One major criticism is on the clarification of the contribution and the overlook of some existing work. There are many recent work on inference-time scaling of (discrete) diffusion models and SMC is classical. The paper would benefit from adding a clarification on the contributions with respect to all these existing work. Another criticism is on the computational cost of the proposed method. A comprehensive study of it could strengthen the paper.

**Reviewer Scores:**

Reviewer jzG5 could have increased score

---

### Decision · Program_Chairs · 2026-01-26

Accept (Poster)